# GlocalCLIP: Object-agnostic Global-Local Prompt Learning for Zero-shot Anomaly Detection

## Abstract

Zero-shot anomaly detection (ZSAD) is crucial for detecting anomalous patterns in target datasets without using training samples, specifically in scenarios where there are distributional differences between the target domain and training data or where data scarcity arises because of restricted access. Although recently pre-trained vision-language models demonstrate strong zero-shot performance across various visual tasks, they focus on learning class semantics, which makes their direct application to ZSAD challenging. To address this scenario, we propose GlocalCLIP, which uniquely separates global and local prompts and jointly optimizes them. This approach enables the object-agnostic glocal semantic prompt to effectively capture general normal and anomalous patterns without dependency on specific objects in the image. We refine the text prompts for more precise adjustments by utilizing deep-text prompt tuning in the text encoder. In the vision encoder, we apply V-V attention layers to capture detailed local image features. Finally, we introduce glocal contrastive learning to improve the complementary learning of global and local prompts, effectively detecting anomalous patterns across various domains. The generalization performance of GlocalCLIP in ZSAD was demonstrated on 15 real-world datasets from both the industrial and medical domains, achieving superior performance compared to existing methods.

## 1 Introduction

Anomaly detection (AD) involves identifying abnormal data that deviate from normal data patterns. It has become a crucial technology in various industries, such as manufacturing and healthcare (Bergmann et al., 2019; 2020; Roth et al., 2022; Xie et al., 2023; Liu et al., 2023a). Traditional AD methods operate through one-class classification that involves learning normal patterns from a single class (Sohn et al., 2020; Zavrtanik et al., 2021; McIntosh & Albu, 2023; Liu et al., 2023c). This approach effectively focuses on learning normal data within a single class; however, its industrial application is severely limited due to the following challenges: (1) A separate model needs to be trained for each class, which is both time-consuming and costly. Additionally, the model requires retraining when a new class is introduced, leading to inefficiency. (2) There may be distributional differences between the training data and the actual test environment data. The discrepancy between the previously learned normal patterns and the target data can degrade the generalization performance of the model, particularly when the target domain has little relevance to the training data. (3) In cases where data access is restricted due to confidentiality, gathering sufficient training data may be difficult, potentially resulting in overfitting or underfitting because of the inability to fully learn normal patterns (Liu et al., 2023b). Recent research has focused on zero-shot anomaly detection (ZSAD) to address these issues. ZSAD enables the detection of anomalous patterns across various classes and domains without relying on training data from the target domain. ZSAD has been effectively applied in various fields owing to the emergence of large-scale pre-trained models, such as vision-language models (VLMs). Among existing VLMs, Contrastive Language-Image Pre-training (CLIP) simultaneously learns images and text, demonstrating strong zero-shot performance in diverse areas, including industrial visual inspection, medical image analysis, video understanding, and robotic vision (Radford et al., 2021; Yao et al., 2021; Tschannen et al., 2023; Geng et al., 2023; Guo et al., 2023; Zhao et al., 2023; Ni et al., 2022; Sontakke et al., 2024). However, CLIP relies heavily

on global information from images, reducing its applicability to ZSAD (Jeong et al., 2023; Chen et al., 2023). To address this issue, Jeong et al. (2023) proposed window-based patches through a multi-scale approach to detect pixel-level anomalies and introduced a compositional prompt ensemble (CPE) to tackle the challenges of finding optimal prompts. Furthermore, Zhou et al. (2023) proposed an object-agnostic prompt that simplifies prompt design by reducing dependency on class semantics and suggested diagonally prominent attention map layer for extracting local features in CLIP. Cao et al. (2024) introduced hybrid prompts for ZSAD by incorporating both static and dynamic learnable prompts into CLIP. Despite these advances, the attempts to learn representations by separating between global and local prompts remain underexplored. As seen in the susceptibility of CLIP to pixel-level detection, it is evident that global and local representation capture slightly different aspects of the object. Motivated by this gap, we focus on an approach inspired by Zhou et al. (2023) to effectively leverage prompt learning while integrating a larger architectural framework. By doing so, we aim to bridge the gap between global and local representation, enabling more robust anomaly detection. In this study, we propose GlocalCLIP, a refine approach designed to overcome the limitations of existing methods by distinctly separating and complementarily learning global and local prompts. Specifically, we design an object-agnostic glocal semantic prompt that applies to both normal and anomalous cases, enabling contextual anomaly detection while explicitly separating global and local prompts. In the text encoder, we utilize deep-text prompt tuning by inserting learnable tokens for fine-grained text prompts. In the vision encoder, we adopt the value-value (V-V) attention mechanism, enabling more precise learning of fine-grained features from local regions (Vaswani, 2017; Zhou et al., 2023; Li et al., 2024). Finally, we propose a glocal contrastive learning to address the insufficient complementarity between independently learned global and local prompts and to jointly optimize their integration. Through experiments on 15 real-world image datasets, GlocalCLIP demonstrates enhanced anomaly detection performance and strong generalization, even in the presence of discrepancies between the training data and the target domain.

The contributions of this paper are summarized as follows.

- We introduce a novel ZSAD approach named GlocalCLIP, a refined framework to explicitly separate global and local prompts through an object-agnostic glocal semantic prompt design. This design enables the learning of prompts that generalize across a wide range of normal and anomalous patterns without being tied to specific object classes, allowing the model to effectively detect fine-grained visual anomalies.

- We address the insufficient complementarity between global and local prompts by introducing a glocal contrastive learning approach. Through joint optimization of global and local prompts, this approach effectively aligns them to capture both global and local visual features, thereby enhancing the robustness of ZSAD.

- Comprehensive experiments validate the effectiveness and generalization capability of GlocalCLIP on 15 real-world datasets, covering a diverse range of classes from both industrial and medical domains, and demonstrate its strong performance and ability to generalize across various categories.

## 2 RELATED WORK

### 2.1 PROMPT LEARNING

Prompt learning has emerged as a key technique to optimize the performance of VLMs for specific scenarios by incorporating carefully designed prompts into input images or text. Initially, prompt engineering utilized static prompt templates to guide VLMs. However, these static prompts often struggle with generalization due to their rigidity and vulnerability to diverse data distributions (Zhou et al., 2022b;a) To mitigate this limitation, methods such as CPE were proposed, combining multiple pre-defined prompts to improve robustness across varied data domains (Jeong et al., 2023). Recent advancements introduced learnable tokens to enable dynamic adaptation in prompt design. Zhou et al. (2022b) proposed the context optimization (CoOp) method, which integrates learnable tokens into text prompt, enhancing the expressiveness of prompts beyond static templates. Furthermore, Li et al. (2024) extended this concept with a semantic concatenation approach, generating multiple negative samples in prompt learning. Notably Zhou et al. (2023) introduced an object-agnostic prompt learning framework that learns generalized features for both normal and anomalous cases

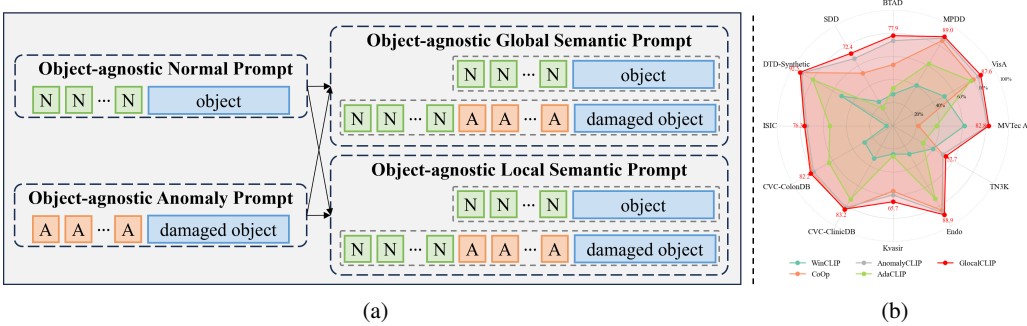

(a)                                                                    (b)

Figure 1: (a) The refinement of prompt design, showing how normal and anomaly prompts are transformed into global and local semantic prompts. (b) Spider chart comparing pixel-level AUPRO scores across different CLIP-based methods on various datasets.

without relying on specific object semantics, as shown in the left part of Fig 1(a). Building upon this foundation, we propose a novel semantic-aware prompt design strategy that transforms these object-agnostic prompts into global and local semantic prompts, as shown in right part of Fig 1(a), enabling more comprehensive feature extraction while maintaining object-agnostic properties.

## 2.2 ZERO-SHOT ANOMALY DETECTION WITH CLIP

CLIP consists of text and vision encoders, where the vision encoder is composed of multilayer networks based on ViT (Dosovitskiy, 2020). For ZSAD, CLIP generates a text embedding $t_c \in \mathbb{R}^D$ by passing a text prompt $\mathbb{T}$, which incorporates the class $c$ from the target class set $C$, through the text encoder. $\mathbb{T}$ follows the format `A photo of a [class]`, where `[class]` represents the target class name. The vision encoder takes an input image $x_i$ and extracts visual features, where the class token $f_i \in \mathbb{R}^D$, referred to as `[cls]` token, is treated as global visual embedding. Additionally, patch token $f_i^m \in \mathbb{R}^{H \times W \times D}$, extracted from detailed regions of the image, are used as local visual embedding. The probability of $x_i$ belonging to class $c$ is calculated based on the cosine similarity between $t_c$ and $f_i$, as shown in the following expression (Zhou et al., 2023):

$$p(y = c|x_i) = P(t_c, f_i) = \frac{exp(< t_c, f_i > /\tau)}{\sum_{c \in \mathcal{C}} exp(< t_c, f_i >)/\tau)}, \tag{1}$$

where $\tau$ represents the temperature hyperparameter and $< \cdot, \cdot >$ represents the cosine similarity. In this study, we assume that object-agnostic prompts are necessary when using CLIP for ZSAD. Under this assumption, we designed two text prompts to distinguish between normal and anomalous conditions and performed anomaly detection by calculating the anomaly probability based on their similarities. Consequently, $t_c$ is represented by two types of text embeddings, where one is the normal text embedding $t_n$ and the other is the abnormal text embedding $t_a$. The anomaly score was denoted as $P(t_a, f_i)$. For local visual embeddings, the probabilities of the normal and anomalous conditions for each pixel $(j, k)$, where $j \in [1, H]$ and $k \in [1, W]$, are calculated as $P(t_n, f_i^{m(j,k)})$ and $P(t_a, f_i^{m(j,k)})$. These probabilities are then used to obtain the normal and anomaly localization maps, $S_n \in \mathbb{R}^{H \times W}$ and $S_a \in \mathbb{R}^{H \times W}$, respectively.

## 3 GLOCALCLIP: OBJECT-AGNOSTIC GLOBAL-LOCAL PROMPT LEARNING

### 3.1 APPROACH OVERVIEW

We proposes the GlocalCLIP, which explicitly separates global and local prompts to learn general normal and anomalous features in a complementary manner. The overall structure is shown in Fig. 2 and comprises four steps: (1) Text Encoder: Prompts from the object-agnostic glocal semantic prompt are passed through the text encoder to generate both global and local text embeddings. (2) Vision Encoder: The input image is processed by the vision encoder, which returns the global and local visual embeddings through the `[CLS]` and patch token, respectively. (3) Anomaly Scoring:

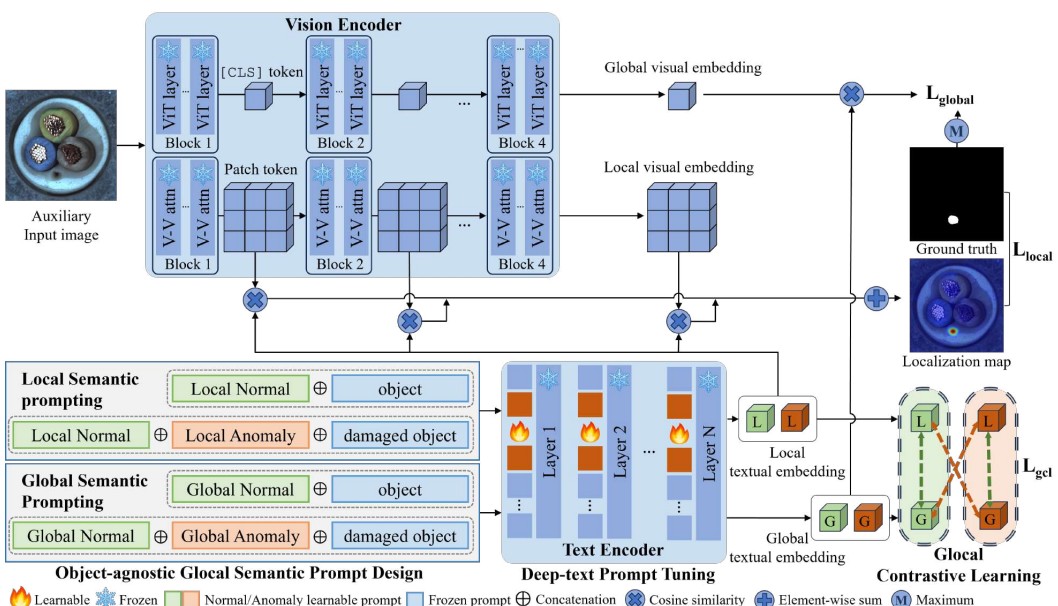

Figure 2: Overview of GlocalCLIP. The object-agnostic glocal semantic prompt enable the text encoder to extract complementary embeddings. Glocal contrastive learning aligns these embeddings to enhance anomaly detection performance. The model optimizes global and local margins to generate anomaly scores and similarity maps, effectively identifying abnormal regions.

Anomaly scores are calculated by measuring the similarity between global visual embedding and global text embedding, as well as between local visual embedding and local text embedding. (4) Glocal Contrastive Learning: Complementary learning is achieved through contrastive learning between global and local text embeddings. During inference, anomaly detection and localization are achieved by computing an anomaly score and generating a localization map.

## 3.2 OBJECT-AGNOSTIC GLOCAL SEMANTIC PROMPT DESIGN

In GlocalCLIP, we propose an object-agnostic glocal semantic prompt design to capture subtle contextual differences between normal and anomalous situations. These prompts are generated by concatenating a suffix indicating an anomaly with a base prompt that represents normal conditions. For example, appending the suffix `with a crack in the corner` to the normal prompt `A crystal clear window` transforms it into an anomaly prompt, `A crystal clear window with a crack in the corner`. Thus, GlocalCLIP can learn both the fine details of an object and its defects through semantic changes in prompts. The text prompts are expressed as follows:

$$p_n = [N_i][object]$$
$$p_a = [N_i][A_j][damaged][object].$$

Normal prompt $p_n$ and anomaly prompt $p_a$ are designed in binary form, following an object-agnostic prompt structure. Here, $[N_i]$ ($i \in [1, E]$) denotes a learnable token for normal conditions, representing the general state of each object. While, $[A_j]$ ($j \in [1, L]$) denotes a learnable token for abnormal conditions, indicating defects or damage specific to each object. Instead of focusing on learning class semantic within an image, these prompts are designed to capture both global and local features that distinguish normal from anomalous conditions. By utilizing the term `object` in a generalized context, the design enables the prompts to learn object-agnostic representations, facilitating a more generalized approach to anomaly detection without reliance on class-specific semantics. In this context, the term `damaged` is manually incorporated into the anomaly prompt to explicitly represent anomalous conditions. By explicitly separating these prompts into global and local contexts, they can be defined as follows:

$$g_n = [N_i^g][object]$$
$$g_a = [N_i^g][A_j^g][damaged][object]$$
$$l_n = [N_i^l][object]$$
$$l_a = [N_i^l][A_j^l][damaged][object].$$

Here, $g_n$ and $g_a$ represent the global prompts for normal and abnormal conditions, respectively, while $l_n$ and $l_a$ correspond to the local prompts for the same conditions. The learnable tokens used in each prompt, $[N_i^g]$, $[N_i^l]$ and $[A_j^g]$, $[A_j^l]$ ($i \in [1, E], j \in [1, L]$), correspond to normal and abnormal states. This separation is designed to learn features from different perspectives in anomaly detection. The global prompts capture the overall context of the image to determine normality or abnormality, while the local prompts focus on fine-grained characteristics of localized defects, such as scratches or contamination. This explicit design enables accurate anomaly detection across various domains by effectively capturing both global and local details. The effect of the learnable prompts when varying their positions is detailed in Appendix C.

### 3.3 GLOCAL CONTRASTIVE LEARNING

To learn between global and local prompts in a complementary manner, we propose a glocal contrastive learning mechanism that aligns text embeddings across different semantic levels. The glocal contrastive learning (GCL) named $L_{gcl}$ operates on triplets of prompts, where global prompt serves as an anchor to regulate its relationship with local prompts. This design choice is motivated by the hierarchical nature of visual understanding: global prompts capture the overall image context, while local prompts refine this understanding with fine-grained details. The loss is formulated as:

$$L_{gcl} = \|\boldsymbol{a} - \boldsymbol{p}\|_2^2 + \max(0, \text{margin} - \|\boldsymbol{a} - \boldsymbol{n}\|_2)^2, \tag{2}$$

where $\boldsymbol{a}$, $\boldsymbol{p}$, and $\boldsymbol{n}$ represent the embeddings of anchor, positive, and negative prompts respectively, and margin determines the minimum required distance between anchor and negative prompts. In GCL, the global normal prompt serves as the anchor, encouraging the local normal prompt to move closer as a positive example, while the local anomaly prompt is pushed farther away as a negative example. Similarly, when the global anomaly prompt is used as the anchor, the local anomaly prompt is brough closer, and the local normal prompt is pushed farther away. This dual alignment ensures that prompts are learned relative to the semantic context of global normality and abnormality. Consequently, the loss function minimizes the distance between semantically similar prompts and maximizes it for dissimilar ones, enabling the model to learn discriminative features where the global context aids the refinement of local details. The total glocal contrastive loss is defined as:

$$L_{gcl}^{total} = L_{gcl}^{normal} + L_{gcl}^{anomaly}, \tag{3}$$

where $L_{gcl}^{normal}$ focuses on aligning prompts under normal conditions, and $L_{gcl}^{anomaly}$ operates on anomaly conditions.

### 3.4 DEEP-TEXT PROMPT TUNING

We utilized deep-text prompt tuning by inserting learnable tokens into each layer of the text encoder, refining text embeddings and enhancing their interaction with visual embeddings. Specifically, at the $i$-th layer, the learnable token $t_i^{learnable}$ is concatenated with the text embedding $t_i$ to adjust the text embedding. This process is represented as follows:

$$t_i' = [t_i^{learnable}, t_i].$$

The updated text embeddings in each layer are passed to the next layer, allowing more detailed text information can be learned using a new $t_i^{learnable}$ token in each layer. Then the text embeddings are aligned with visual features, enabling a more accurate detection of normal and anomaly patterns.

## 3.5 GLOBAL-LOCAL VISUAL FEATURES

In vision encoder, ViT is used to obtain global and local visual embeddings to effectively learn global and local visual features. The original ViT captures a global feature using a `[CLS]` token, while local visual embedding is derived from patch token. In GlocalCLIP, a V-V attention is applied instead of the conventional QKV attention layer to detect fine defects by focusing on local regions. V-V attention replaces both the query and key with the same value, intensifying the correlation among local features. As a result, this modification focus on the fine details of the image, facilitating the detection of subtle anomaly patterns. V-V attention is calculated using

$$Attention(V, V, V) = softmax(VV^T/\sqrt{D})V,$$

where $V$ represents the patch token embedding of the vision encoder, and $D$ denotes the dimension of the visual embedding. The depth at which the V-V attention layer starts to be applied can be adjusted as a hyperparameter to control the focus on local regions.

## 3.6 TRAINING AND INFERENCE

The training loss is composed of three complementary components: global loss $L_{global}$, local loss $L_{local}$ and glocal contrastive loss $L_{gcl}^{total}$ defined in Section 3.3. The global and local loss components in our training objective are inspired by the design principles of Zhou et al. (2023), which effectively balances anomaly detection and localization tasks. The total training loss is defined as:

$$L_{total} = L_{global} + \sum_{M_k \in \mathcal{M}} L_{local}^{M_k} + \lambda L_{gcl}^{total}, \tag{4}$$

where $\mathcal{M}$ denotes a set of intermediate layers used for extracting local features, and $\lambda$ is a hyperparameter controlling interaction between global and local prompt. $L_{global}$ is computed using the binary cross-entropy. Next, $L_{local}$, which is based on the predicted and actual anomaly regions in each measurement, is given by

$$S_{n,M_k}^{(j,k)} = P(l_n, f_{i,M_k}^{m(j,k)}), \ \ S_{a,M_k}^{(j,k)} = P(l_a, f_{i,M_k}^{m(j,k)}), \ \ \text{where} \ j \in [1, H], k \in [1, W]. \tag{5}$$

$S_{n,M_k}^{(j,k)}$ and $S_{a,M_k}^{(j,k)}$ denote the similarities corresponding to the normal and anomalous cases, respectively. Furthermore, let $S \in \mathbb{R}^{H \times W}$ be the ground-truth localization mask, where $S_{j,k} = 1$ denotes that a pixel is anomalous, and $S_{j,k} = 0$ otherwise. The local loss, $L_{local}$, is calculated using

$$L_{local} = Focal(Up([S_{n,M_k}, S_{a,M_k}]), S) + Dice(Up(S_{n,M_k}), I - S) + Dice(Up(S_{a,M_k}), S), \tag{6}$$

where $Focal(\cdot, \cdot)$ and $Dice(\cdot, \cdot)$ represent the loss functions proposed by Ross & Dollár (2017) and Li et al. (2019), respectively. Focal loss assigns higher weights to important samples in imbalanced data, while dice loss is used to reduce the difference between the predicted and actual anomaly regions. $Up(\cdot, \cdot)$ and $[\cdot, \cdot]$ represent upsampling and channel-wise concatenation, respectively, and $I$ denotes a matrix with all elements equal to 1. During inference, anomaly detection and localization are performed based on the anomaly score and localization map, as described in Eq. 1. The anomaly localization map, denoted as $Map \in \mathbb{R}^{H \times W}$, is computed as $Map = G_\sigma(\sum_{M_k \in \mathcal{M}} (\frac{1}{2}(I - Up(S_{n,M_k})) + \frac{1}{2}Up(S_{a,M_k})))$, where $G_\sigma$ represents a Gaussian filter and the parameter $\sigma$ controls the smoothing effect.

## 4 EXPERIMENTS

### 4.1 EXPERIMENT SETUP

**Datasets** To evaluate the performance of the proposed GlocalCLIP model, we conducted experiments on 15 real-world datasets from various industrial and medical domains. For the industrial domains, we used the MVTec AD (Bergmann et al., 2019), VisA (Zou et al., 2022), MPDD (Jezek et al., 2021), BTAD (Mishra et al., 2021), SDD (Tabernik et al., 2020), and DTD-Synthetic (Aota et al., 2023) datasets. In the medical domains, we employed the ISIC (Gutman et al., 2016) dataset for skin cancer detection. The CVC-ClinicDB (Bernal et al., 2015) and CVC-ColonDB (Tajbakhsh et al., 2015) datasets for colon polyp detection. Furthermore, the Kvasir (Jha et al., 2020) and Endo

Table 1: Comparisons of ZSAD performance on industrial domain. The best performance is bold red and the second-best is bold blue. The mean values summarize overall performance across all datasets.

| Task | Category | Datasets | $|\mathcal{C}|$ | CLIP | WinCLIP | CoOp | AnomalyCLIP | AdaCLIP | GlocalCLIP |
|---|---|---|---|---|---|---|---|---|---|
| Image-level (AUROC, AP) | Obj &texture | MVTec AD | 15 | (83.3, 92.4) | (90.4, 95.6) | (82.1, 91.4) | (**91.5**, **96.2**) | (91.2, 95.9) | (**91.7**, **96.4**) |
| | | VisA | 12 | (71.7, 76.6) | (75.6, 78.8) | (77.7, 81.5) | (81.4, **84.9**) | (**81.7**, 84.0) | (**83.7**, **86.2**) |
| | Obj | MPDD | 6 | (71.2, 78.2) | (61.5, 69.2) | (76.0, 78.3) | (**76.9**, **81.4**) | (72.1, 76.0) | (**77.6**, **82.0**) |
| | | BTAD | 3 | (82.7, 86.5) | (68.2, 70.9) | (77.7, 77.7) | (87.5, **90.7**) | (**90.2**, 90.6) | (**89.8**, **92.2**) |
| | | SDD | 1 | (74.0, 57.5) | (84.3, 77.4) | (80.8, 71.0) | (**85.3**, **81.6**) | (81.2, 72.6) | (**86.6**, **84.5**) |
| | Texture | DTD-Synthetic | 12 | (73.7, 89.7) | (95.1, 97.7) | (**96.2**, **98.1**) | (93.7, 97.3) | (**97.8**, **99.0**) | (93.7, 97.3) |
| | | Mean | | (76.1, 80.2) | (79.2, 81.6) | (81.8, 83.0) | (**86.1**, **88.7**) | (85.7, 86.4) | (**87.2**, **89.8**) |
| Pixel-level (AUROC, PRO) | Obj &texture | MVTec AD | 15 | (38.2, 8.8) | (82.3, 61.9) | (44.4, 11.1) | (**91.0**, **81.9**) | (89.4, 37.8) | (**91.4**, **82.8**) |
| | | VisA | 12 | (47.9, 16.1) | (73.2, 51.1) | (42.1, 12.2) | (95.3, **85.1**) | (**95.5**, 77.8) | (**95.9**, **87.5**) |
| | Obj | MPDD | 6 | (42.5, 19.8) | (71.2, 40.5) | (33.7, 14.1) | (96.2, **87.5**) | (**96.4**, 62.2) | (**96.6**, **89.0**) |
| | | BTAD | 3 | (39.5, 7.8) | (72.7, 27.3) | (28.1, 6.5) | (94.5, **73.6**) | (**94.8**, 32.5) | (**96.1**, **77.9**) |
| | | SDD | 1 | (38.7, 10.1) | (68.8, 24.2) | (24.4, 8.3) | (**90.6**, **67.0**) | (71.7, 17.6) | (**93.1**, **72.4**) |
| | Texture | DTD-Synthetic | 12 | (37.6, 15.0) | (79.5, 51.5) | (14.8, 3.0) | (97.8, **91.1**) | (**98.7**, 80.0) | (**98.2**, **92.5**) |
| | | Mean | | (40.7, 12.9) | (74.6, 42.8) | (31.3, 8.5) | (**94.2**, **81.0**) | (91.1, 51.3) | (**95.2**, **83.7**) |

Table 2: Comparisons of ZSAD performance on medical domain. The best performance is bold red and the second-best is bold blue. The mean values summarize overall performance across all datasets. Image-level medical datasets do not provide segmentation ground truth, differentiating them from the pixel-level medical datasets.

| Task | Category | Datasets | $|\mathcal{C}|$ | CLIP | WinCLIP | CoOp | AnomalyCLIP | AdaCLIP | GlocalCLIP |
|---|---|---|---|---|---|---|---|---|---|
| Image-level (AUROC, AP) | Brain | HeadCT | 1 | (84.8, 82.1) | (81.8, 78.9) | (72.1, 74.8) | (**90.8**, **92.2**) | (67.0, 65.0) | (**91.7**, **92.8**) |
| | | BrainMRI | 1 | (88.6, 87.0) | (86.6, 84.1) | (76.9, 78.0) | (**95.4**, **96.0**) | (36.0, 56.2) | (**95.7**, **96.2**) |
| | | Br35H | 1 | (90.2, 85.7) | (80.5, 74.0) | (77.9, 72.8) | (**97.1**, **96.8**) | (42.1, 46.7) | (**97.3**, **97.1**) |
| | | Mean | | (87.9, 84.9) | (83.0, 79.0) | (75.6, 75.2) | (**94.4**, **95.0**) | (48.4, 56.0) | (**94.9**, **95.4**) |
| Pixel-level (AUROC, PRO) | Skin | ISIC | 1 | (65.6, 33.5) | (83.4, 5.5) | (34.5, 2.6) | (**87.4**, **74.5**) | (84.4, 54.5) | (**88.9**, **76.3**) |
| | Colon | CVC-ColonDB | 1 | (52.5, 18.3) | (64.8, 28.4) | (42.3, 3.6) | (**88.5**, **79.3**) | (88.0, 63.9) | (**89.5**, **82.2**) |
| | | CVC-ClinicDB | 1 | (53.0, 25.9) | (70.3, 32.5) | (47.9, 5.4) | (93.0, **81.6**) | (**94.4**, 73.5) | (**93.3**, **84.0**) |
| | | Kvasir | 1 | (45.9, 13.3) | (69.7, 24.5) | (47.7, 7.9) | (93.2, **59.9**) | (**94.6**, 26.2) | (**94.3**, **65.9**) |
| | | Endo | 1 | (42.8, 12.5) | (68.2, 28.3) | (44.0, 5.4) | (94.1, **86.9**) | (**95.2**, 72.8) | (**95.1**, **89.2**) |
| | Thyroid | TN3K | 1 | (39.1, 10.2) | (70.7, 39.8) | (48.6, 4.8) | (**78.2**, **49.7**) | (69.8, 30.0) | (**80.5**, **52.7**) |
| | | Mean | | (49.8, 19.0) | (71.8, 26.5) | (44.2, 5.0) | (**89.1**, **72.0**) | (87.7, 53.5) | (**90.3**, **75.1**) |

(Hicks et al., 2021) datasets were employed for polyp identification, and the TN3k (Gong et al., 2021) dataset was used for thyroid nodule detection. For brain tumor detection, we used HeadCT (Salehi et al., 2021), BrainMRI (Salehi et al., 2021), and Br35H (Hamada., 2020) datasets. More details about the datasets and their analysis can be found in Appendix A.

**Comparison Methods and Evaluation Metrics**  We compared our model with state-of-the-art (SOTA) models, including CLIP (Radford et al., 2021), WinCLIP (Jeong et al., 2023), CoOp (Zhou et al., 2022b), AnomalyCLIP (Zhou et al., 2023), and AdaCLIP (Cao et al., 2024). The evalution was based on the the area under the receiver operating characteristic curve (AUROC) to assess the anomaly detection performance. In addition, for a more detailed analysis, we used the average precision (AP) for anomaly detection precision and AUPRO (Bergmann et al., 2020) to evaluate the anomaly localization accuracy. AUROC indicates how the model distinguishes between normal and abnormal states, AP measures the precision of anomaly detection, and AUPRO evaluates how accurately the model localizes anomalous regions.

**Implementation details**  We adopted the `VIT−L/14@336px` CLIP model[1] as the backbone. All parameters of the CLIP model were kept frozen, and the lengths of the normal and anomaly learnable prompt for both global and local prompts were set to 13 and 10, respectively. The depth of the deep-text prompts was 12 for prompt tuning in the text space, and their length was set to 4. The V-V attention layer was applied at a depth of 6, and multiple patch tokens were used, evenly drawn from the outputs of layers 6, 12, 18, 24. For training the GlocalCLIP, we used the MVTec AD dataset for the industrial domain and the Clinic DB for the medical domain. After training, we evaluated the performance on different datasets. For the MVTec AD, we trained the model using the VisA test data, and for the CVC-Clinic DB, we trained the model using the CVC-Colon DB. To ensure equal comparison, all benchmark models were trained and evaluated using the same setting, and results were reported at the dataset level by averaging performances across each sub-dataset. All experiments were conducted on a single NVIDIA RTX 4090 24 GB GPU. More details can be found in Appendix B.

---

[1] https://github.com/mlfoundations/open_clip

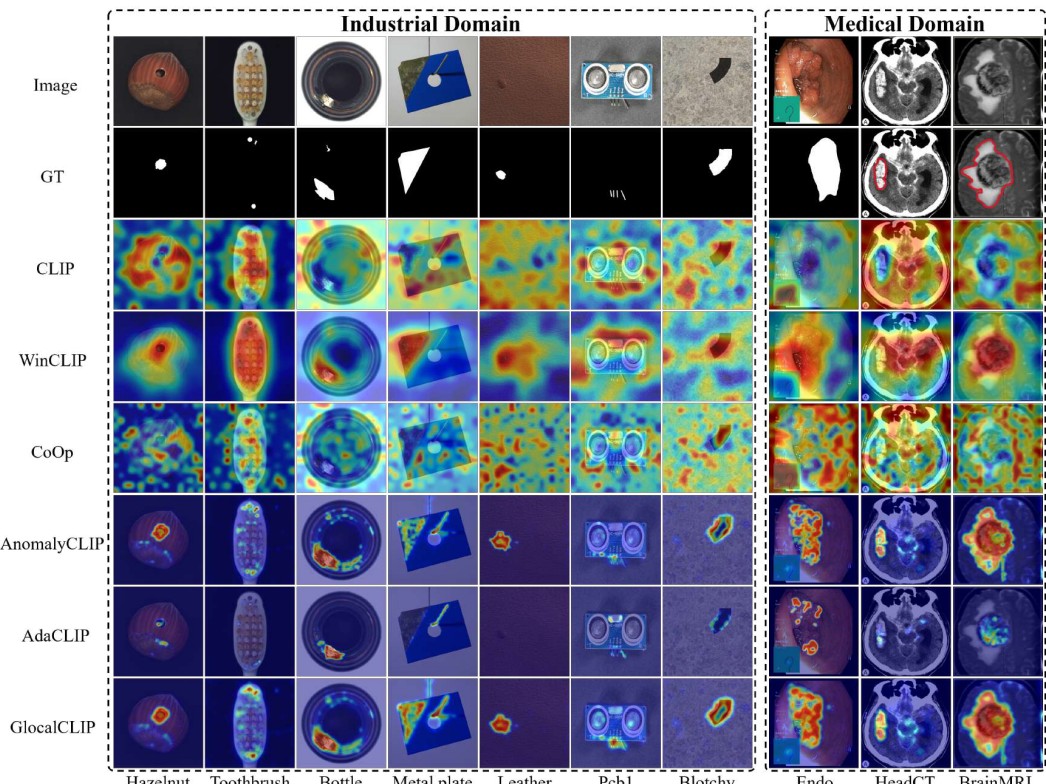

Figure 3: Comparison of ZSAD results across industrial and medical domains. The first row displays input images from the industrial domain (Hazelnut, Bottle, Metal plate, Leather, Pcb1, Blotchy, and Electrical commutators) and the medical domain (HeadCT, BrainMRI, Endo). The second row presents the ground truth anomaly regions for each image. The remaining rows show the anomaly heatmaps generated by different models: CLIP, WinCLIP, CoOp, AnomalyCLIP, AdaCLIP, and GlocalCLIP.

## 4.2 MAIN RESULTS

**Quantitative comparison** As shown in Table 1, GlocalCLIP demonstrated superior ZSAD performance across six industrial datasets, including diverse objects, backgrounds, and anomaly types. Since CLIP and CoOp focus on object class semantics, their performance is inferior for anomaly localization. On the other hand WinCLIP shown better pixel-level performance than CLIP and CoOp by utilizing multi-scale window patches and CPE. AnomalyCLIP achieved advanced performance in both image- and pixel-level tasks through its object-agnostic prompt and specialized architecture. Similarly, AdaCLIP exhibited slightly lower or comparable scores. Our method, GlocalCLIP, demonstrated superior performance by employing a simple yet effective glocal semantic prompt. Compared to AnomalyCLIP, advantage of GlocalCLIP lies in leveraging global and local prompts that learn slightly different representations and complement each other effectively, thereby enhancing the understanding of normal and anomalous patterns. The generalization performance of GlocalCLIP in medical domain was evaluated using nine different datasets, as shown in Table 2. GlocalCLIP achieved the best performance on the HeadCT, BrainMRI, Br35H, ISIC, CVC-ColonDB, and TN3K datasets. Additionally, it ranked first or second on the remaining datasets and first on mean score. These results highlight the effectiveness of glocal semantic prompting in delivering generalization capabilities for ZSAD across both the industrial and medical domains.

**Qualitative comparison** Fig. 3 shows a comparison of anomaly localization maps across the test domain datasets. In the industrial domain, images containing various defect types, such as hazelnuts, toothbrushes, bottles, metal plates, leather, pcb1, and blotchy. CLIP, CoOp, and WinCLIP struggle to capture fine-grained local anomaly regions. CLIP misinterprets normal and anomalous regions, demonstrating the need for adjustment in ZSAD applications. AnomalyCLIP demonstrated

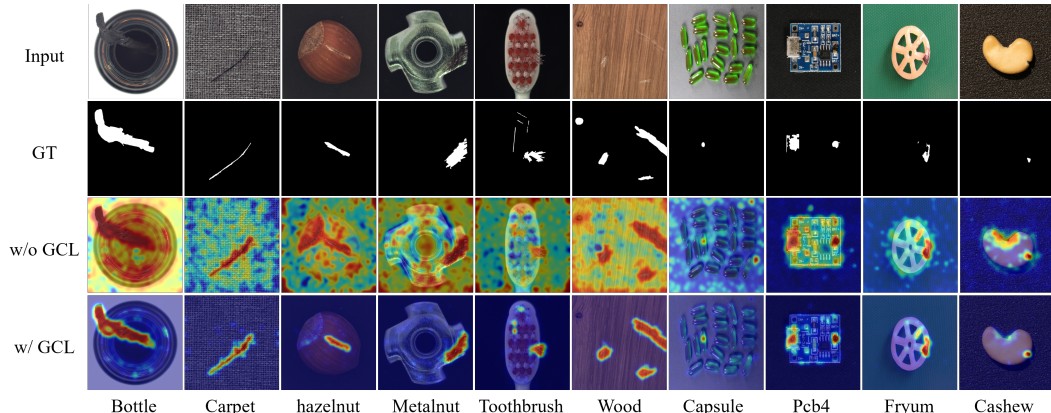

|  | Bottle | Carpet | hazelnut | Metalnut | Toothbrush | Wood | Capsule | Pcb4 | Fryum | Cashew |

Figure 4: Visualization of anomaly localization maps using global prompts with and without GCL. The first row shows sample images from the industrial domain, and the second row provides the true anomaly regions. The third row displays localization maps generated without GCL, where the global prompt struggles to precisely localize pixel-level anomalies. The last row shows localization maps generated with GCL, where the model demonstrates improved detection of both global and local anomalies, effectively localizing fine-grained anomalous regions.

Table 3: Module ablation

| Module | Industrial domain | | Medical domain | |
|---|---|---|---|---|
|  | Pixel-level | Image-level | Pixel-level | Image-level |
| Base | (33.4, 9.0) | (80.3, 82.5) | (47.1, 25.0) | (89.3, 88.9) |
| $+F_1$ | (92.2, 80.0) | (80.6, 83.0) | (73.1, 46.6) | (89.5, 89.2) |
| $+F_2$ | (**95.0**, 82.4) | (85.6, 88.3) | (90.0, **74.5**) | (**94.6**, **95.2**) |
| $+F_3$ | (**95.3**, **84.0**) | (**86.2**, **88.5**) | (**90.2**, 74.4) | (89.8, 91.1) |
| $+F_4$ | (**95.3**, **83.3**) | (**86.7**, **89.3**) | (**90.3**, **74.8**) | (**94.9**, **95.4**) |

Table 4: Prompt design ablation

| Prompt type | Semantic design | Industrial domain | | Medical domain | |
|---|---|---|---|---|---|
|  |  | Pixel-level | Image-level | Pixel-level | Image-level |
| Single | ✗ | (94.2, 81.0) | (86.0, 88.7) | (89.1, 72.0) | (94.4, 95.0) |
|  | ✓ | (**94.3**, 81.2) | (85.9, 88.5) | (89.2, 71.7) | (94.5, 95.0) |
| Glocal | ✗ | (**95.2**, **83.5**) | (**86.8**, **89.3**) | (**90.2**, **74.6**) | (**95.0**, **95.5**) |
|  | ✓ | (**95.2**, **83.7**) | (**87.2**, **89.8**) | (**90.3**, **75.2**) | (**95.2**, **95.6**) |

reasonable performance; however, it occasionally failed to capture certain anomaly regions that required a broader global perspective. In the medical domain, visualization results from the HeadCT, BrainMRI, and Endo datasets. CLIP and CoOp faced difficulties detecting anomalies, and while AdaCLIP performed well in certain cases, it failed to fully capture defects in some medical images. The explicit separation of global and local prompts in GlocalCLIP enables it to learn the distribution of normal and anomalous samples independently, and then enhance complementary learning, resolving the trade-off between image- and pixel-level performances caused by a lack of complementary information. Consequently, GlocalCLIP achieves the best ZSAD performance across both industrial and medical domains, demonstrating its generalization capability.

## 4.3 ABLATION STUDY

**Module ablation** We conducted a series of module comparison experiments to demonstrate the effectiveness of the key components of GlocalCLIP by evaluating the performance impact of each major module through module addition. Table 3 presents the comparison, where the base model is the standard CLIP. The modules are as follows: $F_1$ is V-V attention with multilayer structure; $F_2$ involves semantic prompt design with deep-text prompt tuning; $F_3$ separates global and local prompts; and $F_4$ applies glocal contrastive loss for complementary learning. The prompts of the base model were set to A photo of a [class] and A photo of a damaged [class], While the baseline performed well with global visual information, it lacked the precision required to detect local anomalies. Adding $F_1$ significantly improved detection performance for local regions. $F_2$ allowed for a more precise anomaly detection through prompt learning. $F_3$ enhanced the performance by separately learning the global and local information. Finally, $F_4$ improved the generalization by enabling complementary learning between global and local embeddings. These results confirmed that each component played a critical role in improving ZSAD performance by supporting the learning of both global and local information.

**Prompt design ablation** We evaluated the effect of object-agnostic glocal semantic prompt design settings. Table 4 presents comparisons across different prompt types and semantic designs.

The results show that the glocal semantic prompt design enables more accurate learning of diverse visual patterns between normal and abnormal samples, leading to improved anomaly detection performance. Specifically, the glocal prompt design consistently outperforms the single prompt design across both industrial and medical domains at the pixel and image levels. Additionally, semantic prompt design outperforms the default setting.

**Reverse global-local prompts with glocal contrastive learning**   We visualized glocal contrastive learning (GCL) on global and local prompts through visual comparisons, as shown in Fig. 4. The figure presents pixel-level anomaly localization maps with and without GCL. Specifically, w/ GCL shows localization maps generated from global prompts trained with GCL, while w/o GCL depicts results without GCL. Without GCL, the maps capture some local features, reflecting an understanding of local anomalies, but mainly focus on the overall image, resulting in less precise localization. In contrast, GCL incorporates local information, enabling complementary learning between global and local features. Balancing global and local performance was challenging during experiments, as enhancing one often came at the expense of the other. This challenge motivated the separation of global and local prompts during training, followed by GCL integration to unify their complementary strengths. As a result, prompts trained with GCL improved anomaly detection and localization by effectively capturing features at both global and local levels. These findings highlight the complementary nature of global and local prompts in understanding and localizing anomalies.

## 5   CONCLUSION

In this study, we propose a novel ZSAD approach named GlocalCLIP, which detects anomalies through the unique strategy of explicitly separating global and local prompts. By training these prompts in a complementary manner, GlocalCLIP effectively captures fine-grained features. Prompts trained using object-agnostic glocal semantic prompt design and glocal contrastive learning demonstrated strong generalization performance across various domains, achieving impressive results in both the medical and industrial sectors. Experimental results from 15 diverse image datasets confirmed that GlocalCLIP outperforms SOTA models in ZSAD and surpasses existing CLIP-based models. While this study focused on visual anomaly detection, expanding the method to accommodate a wider range of anomaly scenarios, including logical errors, is necessary. Future research should address approaches to bridge the modality gap between images and text. The novel perspective introduced by GlocalCLIP is expected to contribute significantly to advancements in this field.

### REPRODUCIBILITY STATEMENT

To ensure reproducibility, our Appendix includes five main sections. Appendix A outlines dataset statistics, while Appendix B details the implementation of GlocalCLIP and baseline reproduction. Appendix C covers additional ablations on hyperparameters and the effect of anomaly prompt. Appendices D and E provide visualizations and ZSAD performance on data subsets.

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

## A  DATASETS

**Dataset overview**    In this study, we evaluated the performance of GlocalCLIP on 15 public datasets from both industrial and medical domains. The test sets of each dataset were used for validation, with detailed dataset-specific information provided in Table 5. To ensure consistency, standard normalization techniques from OpenCLIP were applied across all datasets, and image sizes were standardized to a resolution of (518, 518) for uniform visual feature map resolution.

**Industrial dataset analysis**    In this study, we utilized datasets that include a variety of objects and texture-based anomalies. A key distinction between MVTec AD and VisA lies in the composition of the objects within their images. MVTec AD primarily consists of single-object images, where each image focuses on a single object and its potential defects. In contrast, VisA often includes images containing multiple instances of objects from the same class (e.g., candles, capsules, macaroni), as well as defects that overlap between objects (e.g., cashew, fryum, pipe fryum). This multi-object setting in VisA increases the complexity of anomaly detection, as defects may be localized to only one of several objects. When training on MVTec AD and testing on VisA, performance improved by separating global and local prompts, as a single prompt is less effective at identifying anomalies spread across multiple objects. In such complex scenarios, distinguishing between global and local prompts allows for more precise detection of localized defects.

## B  IMPLEMENTATION DETAILS AND BASELINE METHODS

### B.1  IMPLEMENTATION DETAILS

In this study, we adopted the `VIT-L/14@336px` CLIP model as the backbone, kept all parameters frozen. Across all datasets, the normal prompt length was set to 13, the anomaly prompt length to 10, and the deep-text prompt length and depth to 4 and 12, respectively. The margin was set to 0, lambda to 1, and the Gaussian filter size $\sigma$ to 8, except in specific cases. For MVTec AD, the abnormal suffix length was set to 13, and the depth was set to 9. Additionally, for VisA, BTAD, and SDD, lambda was set to 0. In the medical datasets, the deep-text prompt length was set to 2

Table 5: Key statistics on the datasets used.

| Domain | Dataset | Category | Modalities | $|\mathcal{C}|$ | Normal and anomalous samples |
|---|---|---|---|---|---|
| Industrial | MVTec AD | Obj &texture | Photography | 15 | (467, 1258) |
| | VisA | | Photography | 12 | (962, 1200) |
| | MPDD | Obj | Photography | 6 | (176, 282) |
| | BTAD | | Photography | 3 | (451, 290) |
| | SDD | | Photography | 1 | (181, 74) |
| | DTD-Synthetic | Texture | Photography | 12 | (357, 947) |
| Medical | ISIC | Skin | Photography | 1 | (0, 379) |
| | CVC-ClinicDB | | Endoscopy | 1 | (0, 612) |
| | CVC-ColonDB | Colon | Endoscopy | 1 | (0, 380) |
| | Kvasir | | Endoscopy | 1 | (0, 1000) |
| | Endo | | Endoscopy | 1 | (0, 200) |
| | TN3K | Thyroid | Radiology (Utralsound) | 1 | (0, 614) |
| | HeadCT | | Radiology (CT) | 1 | (100, 100) |
| | BrainMRI | Brain | Radiology (MRI) | 1 | (98, 155) |
| | Br35H | | Radiology (MRI) | 1 | (1500, 1500) |

for Colon DB. For HeadCT, Kvasir, and Endo, a deep-text prompt length of 2 provided the best performance. Furthermore, for Br35h and Brain MRI, a lambda value of 0.01 was optimal, while for Th3k, a lambda of 0.1 performed better. The training epoch was set to 15, and the learning rate to 0.001, using the adam optimizer with $\beta_1$ and $\beta_2$ set to 0.5 and 0.999, respectively. All experiments were conducted on PyTorch-2.0.0 with a single NVIDIA RTX 4090 GPU.

## B.2 BASELINE METHODS

To demonstrate the superiority of GlocalCLIP, we compared its performance with several SOTA models in ZSAD. The details of the implementations and reproductions for each baseline method are as follows:

- CLIP (Radford et al., 2021). CLIP is a vision-language model that learns to associate images with corresponding text descriptions through contrastive learning. We employ text prompt templates for ZSAD as `A photo of a normal [class]` and `A photo of a damaged [class]`, where `[class]` denotes the target class name. The anomaly score is computed according to Eq. 1. For anomaly localization, this computation is extended to local visual embeddings. All parameters were kept the same as specified in their paper.

- WinCLIP (Jeong et al., 2023). WinCLIP is a SOTA model for ZSAD that applies a window-based approach to CLIP to enhance anomaly detection. Additionally, they propose a compositional prompt ensemble by utilizing a large number of prompt templates. All parameters were kept the same as specified in their paper.

- CoOp (Zhou et al., 2022b). CoOp is a context optimization method for vision-language models that learns optimal prompts to improve performance across various downstream tasks. We used a text prompt design as $[V_1][V_2]...[V_N]$`[class]` and $[W_1][W_2]...[W_N]$`[damaged][class]` for equal comparison on ZSAD. All parameters were kept the same as specified in their paper.

- AnomalyCLIP (Zhou et al., 2023). AnomalyCLIP is a SOTA model for ZSAD that introduces object-agnostic prompt design. Additionally, they propose DPAM, which uses V-V attention for accurate localization of anomalous regions. All parameters are kept the same as specified in their paper.

- AdaCLIP (Cao et al., 2024). AdaCLIP is a SOTA model for ZSAD model that introduces hybrid prompts, combining both static and dynamic prompts. All parameters were kept the same as specified in their paper.

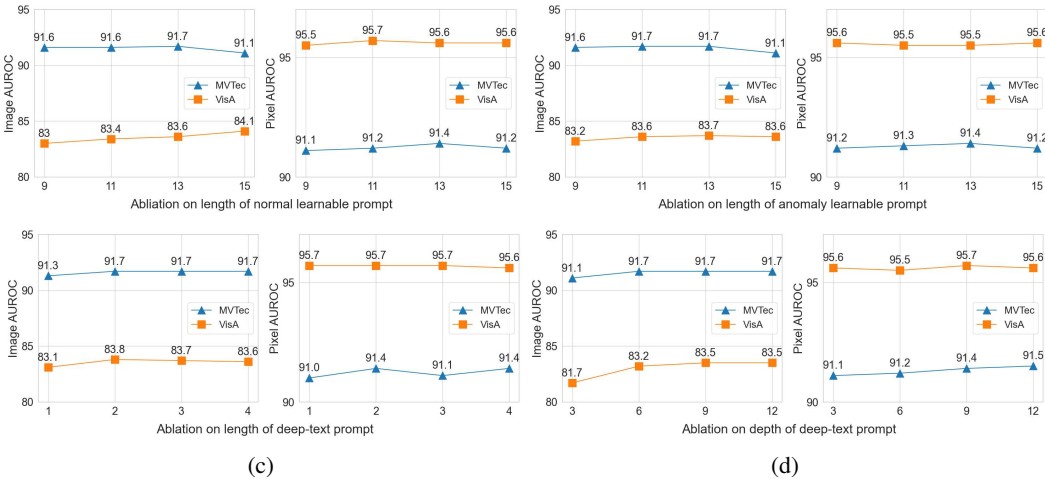

Figure 5: Hyperparameter analysis. (a) is the length of normal learnable prompt, (b) is the length of anomaly learnable prompt, (c) is the length of deep-text prompts, and (d) is the depth of deep-text prompts. Image- and pixel-level performance (AUROC, AUROC) is reported for MVTec AD and VisA datasets, presented on the left and right sides of each subplot, respectively.

## C   ADDITIONAL ABLATIONS

**Hyperparameter ablation**   The results of the hyperparameter experiments are presented in Fig. 5, where we evaluate performance variations across four key hyperparameters. Fig. 5a illustrates that a normal learnable prompt length of 13 provides robust performance. Similarly, as shown in Fig. 5b, an anomaly prompt length of 13 is found to be optimal. Fig. 5c indicates that setting the length of the deep-text prompt to 4 achieves optimal results. Lastly, Fig. 5d demonstrates the effect of varying the depth of the deep-text prompt layers in the text encoder, with an optimal depth of 12 layers maximizing performance, while a shallower depth leads to reduced performance.

**Module ablation based on AnomalyCLIP**   To assess the effectiveness of the proposed approach compared to AnomalyCLIP, we conducted an ablation study, as summarized in Table 6. The results demonstrate that incorporating each module progressively improves performance. Adding the Semantic design module led to notable enhancements, with pixel-level performance increasing from 94.2 to 95.0 in the industrial domain and from 89.1 to 90.0 in the medical domain. Furthermore, the addition of the Global-local branch and GCL modules resulted in even greater performance gains. This combination achieved the highest performance across all domains and evaluation metrics, highlighting the ability of the Global-local branch and GCL to facilitate more precise feature learning.

Table 6: Module ablation based on AnomalyCLIP

| Module | Industrial domain | | Medical domain | |
|---|---|---|---|---|
| | Pixel-level | Image-level | Pixel-level | Image-level |
| AnomalyCLIP | (94.2, 81.0) | (86.1, 88.7) | (89.1, 72.0) | (94.4, 95.0) |
| + Semantic design | (95.0, 82.4) | (85.6, 88.3) | (90.0, 74.5) | (94.6, 95.2) |
| + Global-local branch | (95.3, 84.0) | (86.2, 88.5) | (90.2, 74.4) | (89.8, 91.1) |
| + GCL | (95.3, 83.3) | (86.7, 89.3) | (90.3, 74.8) | (94.9, 95.4) |

**Anchor prompt ablation**   To demonstrate the effectiveness of the anchor prompt design, which incorporates hierarchical semantics, we conducted an ablation study summarized in Table 7. The results clearly indicate that the global prompt consistently outperforms the local prompt across all domains and evaluation metrics. These findings demonstrate that using a global prompt as the anchor enables a more comprehensive representation of features, leading to better overall performance.

Table 7: Anchor prompt ablation

| Anchor prompt | Industrial domain | | Medical domain | |
|---|---|---|---|---|
| | Pixel-level | Image-level | Pixel-level | Image-level |
| Local prompt | (95.1, 83.0) | (86.6, 89.0) | (90.2, 74.8) | (94.8, 95.3) |
| Global prompt | (95.3, 83.3) | (86.7, 89.3) | (90.3, 74.8) | (94.9, 95.4) |

Table 9: Comparison of normal and anomaly prompt positions in ZSAD performance within the industrial domain.

| Task | Category | Datasets | $|\mathcal{C}|$ | $[N][obj][A]$ | $[A][N][obj]$ | $[N][A][obj]$ |
|---|---|---|---|---|---|---|
| | Obj &texture | MVTec AD | 15 | (91.6, 96.3) | (91.6, 96.3) | (91.7, 96.4) |
| | | VisA | 12 | (83.5, 86.0) | (83.7, 86.2) | (83.7, 86.2) |
| Image-level | Obj | MPDD | 6 | (77.7, 82.1) | (78.0, 82.2) | (77.6, 82.0) |
| (AUROC, AP) | | BTAD | 3 | (88.9, 90.7) | (89.0, 92.3) | (89.8, 92.2) |
| | | SDD | 1 | (86.5, 83.4) | (86.7, 84.7) | (86.6, 84.5) |
| | Texture | DTD-Synthetic | 12 | (93.7, 97.3) | (93.7, 97.3) | (93.7, 97.3) |
| | | Mean | | (87.0, 89.3) | (87.0, 89.8) | (87.2, 89.8) |
| | Obj &texture | MVTec AD | 15 | (91.4, 82.5) | (97.6, 83.3) | (91.4, 82.8) |
| | | VisA | 12 | (95.9, 87.6) | (95.9, 87.4) | (95.9, 87.5) |
| Pixel-level | Obj | MPDD | 6 | (96.5, 88.6) | (96.6, 89.1) | (96.6, 89.0) |
| (AUROC, PRO) | | BTAD | 3 | (96.1, 78.9) | (96.1, 78.4) | (96.1, 77.9) |
| | | SDD | 1 | (93.3, 71.0) | (93.2, 71.2) | (93.1, 72.4) |
| | Texture | DTD-Synthetic | 12 | (98.2, 92.2) | (98.2, 92.4) | (98.2, 92.5) |
| | | Mean | | (95.2, 83.5) | (95.3, 83.6) | (95.2, 83.7) |

Table 10: Comparison of normal and anomaly prompt positions in ZSAD performance within the medical domain.

| Task | Category | Datasets | $|\mathcal{C}|$ | $[N][obj][A]$ | $[A][N][obj]$ | $[N][A][obj]$ |
|---|---|---|---|---|---|---|
| | | HeadCT | 1 | (91.8, 93.1) | (92.3, 93.5) | (91.7, 92.8) |
| Image-level | Brain | BrainMRI | 1 | (95.8, 96.2) | (95.8, 96.2) | (95.7, 96.2) |
| (AUROC, AP) | | Br35H | 1 | (97.4, 97.2) | (97.3, 97.1) | (97.3, 97.1) |
| | | Mean | | (95.0, 95.5) | (95.1, 95.6) | (94.9, 95.4) |
| | Skin | ISIC | 1 | (89.0, 78.1) | (89.0, 75.8) | (88.9, 76.3) |
| | | CVC-ColonDB | 1 | (89.4, 82.0) | (89.5, 82.0) | (89.5, 82.2) |
| Pixel-level | Colon | CVC-ClinicDB | 1 | (93.4, 83.1) | (93.4, 83.3) | (93.3, 84.0) |
| (AUROC, PRO) | | Kvasir | 1 | (94.2, 65.7) | (94.2, 65.6) | (94.3, 65.9) |
| | | Endo | 1 | (95.0, 88.8) | (95.0, 88.8) | (95.1, 89.2) |
| | Thyroid | TN3K | 1 | (80.3, 52.4) | (80.1, 51.7) | (80.5, 52.7) |
| | | Mean | | (90.2, 75.0) | (90.2, 76.2) | (90.3, 75.1) |

**Glocal contrastive learning**   In this study, we evaluated the impact of the hyperparameter lambda on the trade-off between global and local performance in ZSAD. We observed that increasing the emphasis on global information, such as through higher global loss penalties or additional networks designed to enhance global features, often resulted in decreased local performance. This finding underscores the importance of balancing global and local representations in ZSAD.

To address this, we propose glocal contrastive learning, which aims to integrate global and local information in a complementary manner. Notably, the mere design of separate global and local prompts led to performance that surpassed SOTA methods, and the complementary learning framework further enhanced the generalization ability of ZSAD. While, as discussed in Appendix A, pixel-level performance occasion-

Table 8: Lambda ablation

| Lambda | Industrial domain | | Medical domain | |
|---|---|---|---|---|
| | Pixel-level | Image-level | Pixel-level | Image-level |
| 0.0 | (95.3, 84.0) | (86.2, 88.5) | (90.2, 74.4) | (89.8, 91.1) |
| 0.001 | (95.3, 83.8) | (86.2, 88.6) | (90.3, 74.5) | (93.9, 94.6) |
| 0.01 | (95.3, 83.6) | (86.5, 88.9) | (90.2, 74.4) | (95.2, 95.6) |
| 0.1 | (95.3, 83.3) | (86.8, 89.0) | (90.2, 74.6) | (93.6, 90.6) |
| 1 | (95.3, 83.3) | (86.7, 89.3) | (90.3, 74.8) | (94.8, 95.4) |

ally showed better results in specific industrial contexts, the glocal framework consistently demonstrated robust and balanced anomaly detection across various domains. The ablation study on the hyperparameter lambda is presented in Table 6, providing further insights into its effect on model performance.

**Positioin of anomaly prompt**   We conducted experiments to examine the performance change based on the position of the abnormal learnable token. Tables 7 and 8 demonstrate that positioning the learnable token at the front, e.g., $[N][A][\texttt{object}]$ or $[A][N][\texttt{object}]$, leads to improved performance. This is because a token placed at the beginning of a sentence often sets the context and introduces the topic, thereby making it more influential for the model. Therefore, optimizing the position of tokens is important for model performance.

# D   VISUALIZATION

**Global and local prompt visualization**   To illustrate the distinction between global and local prompts, we visualized their embeddings within the latent space for the class in the VisA dataset, as

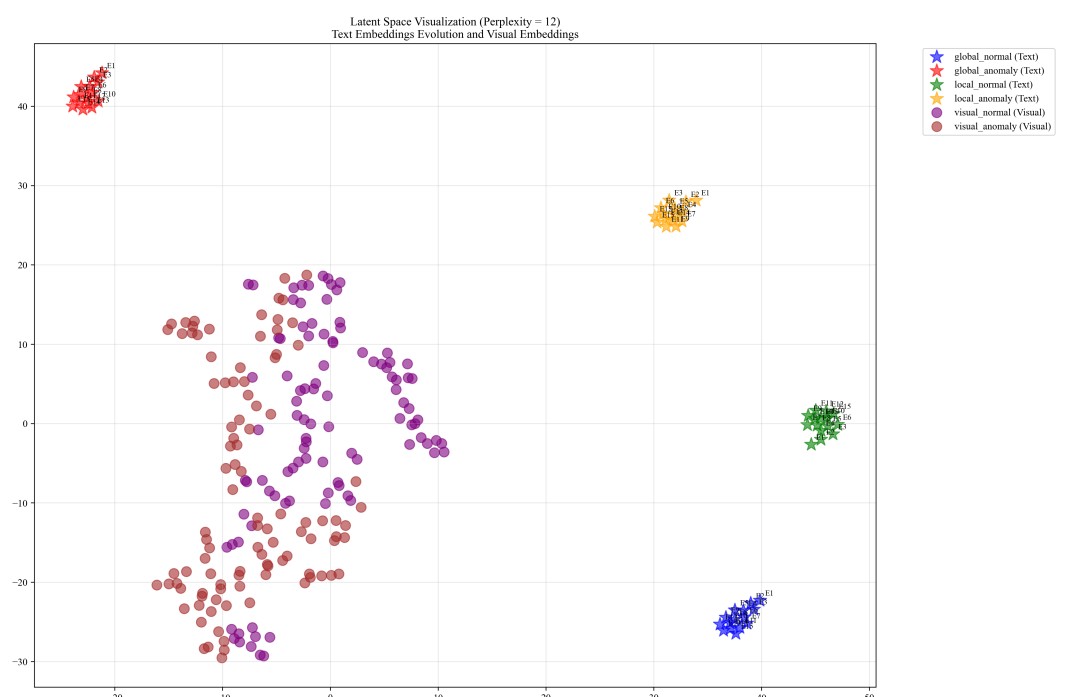

Figure 6: Visualization of global and local prompt for candles class within the VisA dataset.

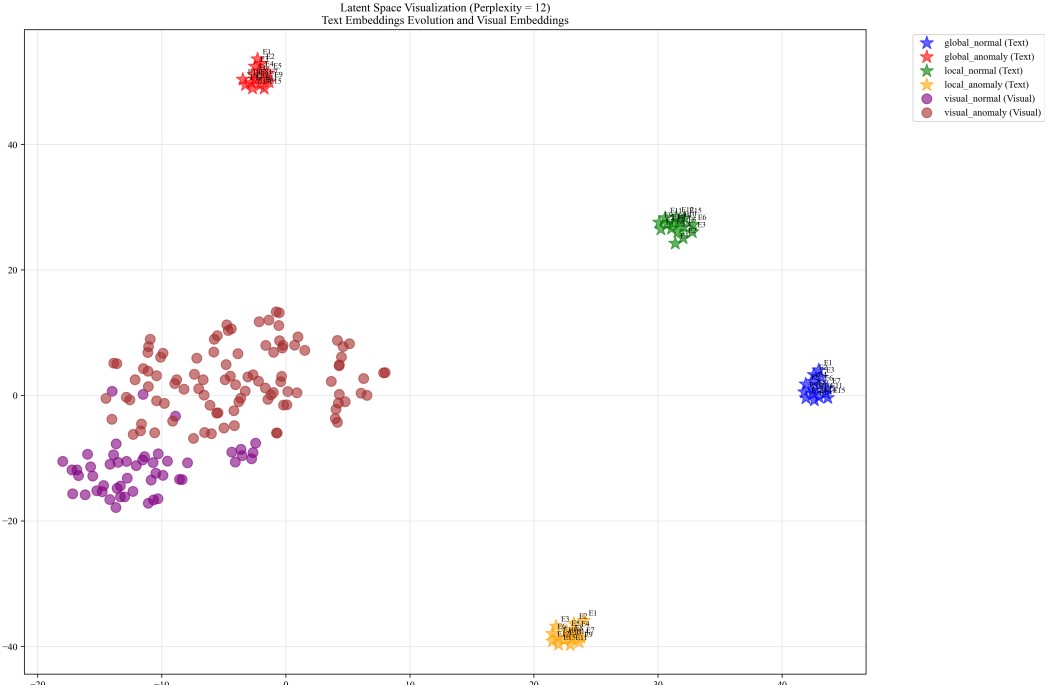

Figure 7: Visualization of global and local prompt for cashew class within the VisA dataset.

shown in Fig. 6 to 8. In these figures, E1 to E15 represent the prompts at each epoch, reflecting the progression of learning within the latent space.

**Similarity score between textual and visual embeddings.** We visualized the similarity scores between the textual prompts and visual embeddings across different datasets. By comparing these similarity scores, we can observe how effectively the prompts align with visual features and identify

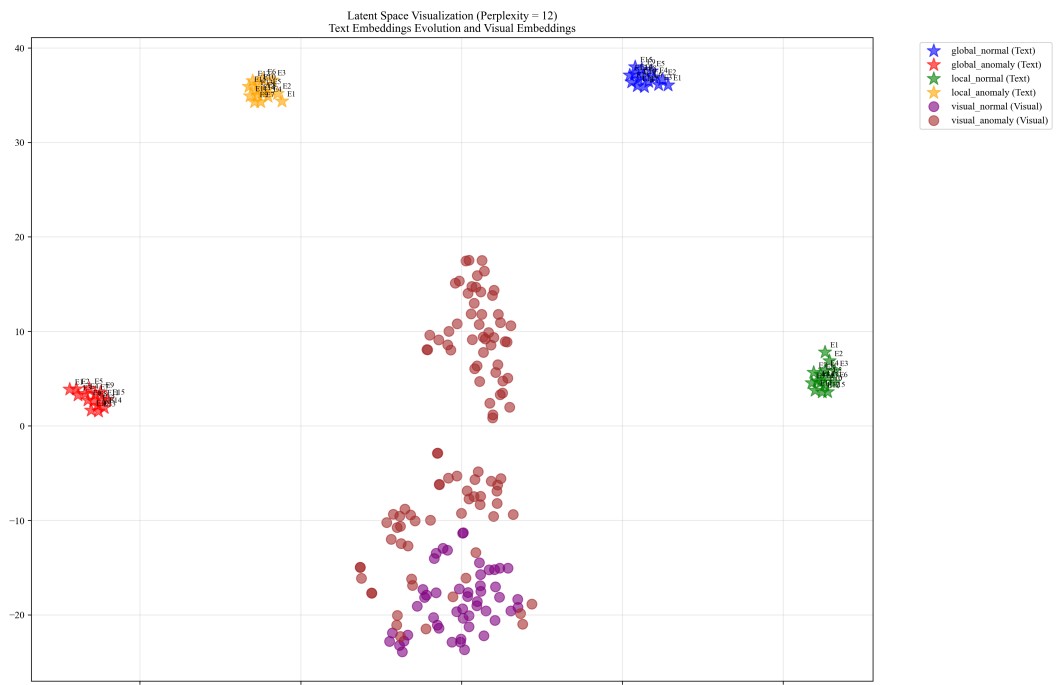

Figure 8: Visualization of global and local prompt for chewinggum class within the VisA dataset.

patterns that differentiate normal and anomalous samples. The visualization provides insights into the degree of alignment between textual and visual representations, offering a deeper understanding performance of the model in associating text-based descriptions with visual content. Specifically, Fig. 6 shows the results for the MVTec AD dataset, Fig. 7 shows the results for the VisA dataset, and Fig. 8 shows the results for medical domain datasets.

**Anomaly localization map for different datasets.** We provide visualizations of the anomaly score maps for various datasets to illustrate how anomalies are detected and localized. These score maps highlight regions of the input images that exhibit abnormal features, as determined by the model. By examining the distribution and intensity of the anomaly scores, we can gain a clearer understanding of how the model identifies and differentiates anomalies from normal patterns across diverse datasets. This visualization serves to showcase the effectiveness of the model in detecting anomalies within different contexts and domains. Specifically, Figs. 9 to 20 show results for various categories in the MVTec AD dataset, including hazelnut, capsule, carpet, pill, screw, leather, wood, metal nut, grid, zipper, and tile. Figs. 21 to 23 present results for metal plate, tubes, and white bracket from the MPDD dataset. Fig. 24 shows the anomaly localization for blotchy in the DTD-synthetic dataset. Figs. 25 to 29 display results for cashew, candle, pipe fryum, chewing gum, and capsules in the VisA dataset. Figs. 30 and 31 illustrate skin anomalies in ISIC and thyroid anomalies in TN3K, respectively. Fig. 32 presents colon anomalies in CVC-ColonDB, and finally, Fig. 33 visualizes brain anomalies in BrainMRI.

## E    FINE-GRAINED ZSAD RESULTS

In this section, we present the fine-grained data subset-level ZSAD performance in details.

Table 11: Fine-grained comparison experiment of anomaly localization performance (AUROC) across data classes in MVTec AD.

| Object name | CLIP | WinCLIP | CoOp | AnomalyCLIP | AdaCLIP | GlocalCLIP |
|---|---|---|---|---|---|---|
| Carpet | 23.3 | 90.9 | 37.4 | 98.9 | 96.9 | 99 |
| Bottle | 23.7 | 85.7 | 33.9 | 90.2 | 91.0 | 90.8 |
| Hazelnut | 44.4 | 95.7 | 57.3 | 97.3 | 97.7 | 97.4 |
| Leather | 6.6 | 95.5 | 41.1 | 98.7 | 99.4 | 98.8 |
| Cable | 48.1 | 61.3 | 59.2 | 79.1 | 74.7 | 79.6 |
| Capsule | 58.7 | 87.0 | 51.9 | 95.9 | 94.3 | 95.9 |
| Grid | 11.0 | 79.4 | 26.6 | 97.3 | 94.3 | 97.3 |
| Pill | 60.3 | 72.7 | 49.8 | 91.2 | 88.0 | 92.5 |
| Transistor | 40.6 | 83.7 | 46.9 | 70.5 | 60.0 | 72.7 |
| Metal_nut | 33.1 | 49.3 | 48.8 | 75.4 | 70.9 | 71.9 |
| Screw | 65.5 | 91.1 | 36.5 | 97.4 | 97.8 | 97.8 |
| Toothbrush | 54.7 | 86.2 | 62.3 | 91.2 | 97.5 | 92.0 |
| Zipper | 40.2 | 91.7 | 17.8 | 90.6 | 95.3 | 92.5 |
| Tile | 41.5 | 79.1 | 44.5 | 94.8 | 88.5 | 95.6 |
| Wood | 21.8 | 85.1 | 52 | 96.5 | 94.3 | 96.8 |
| Mean | 38.2 | 82.3 | 44.4 | 91.0 | 89.4 | 91.4 |

Table 12: Fine-grained comparison experiment of anomaly localization performance (PRO) across data classes in MVTec AD.

| Object name | CLIP | WinCLIP | CoOp | AnomalyCLIP | AdaCLIP | GlocalCLIP |
|---|---|---|---|---|---|---|
| Carpet | 8.8 | 66.3 | 10.6 | 90.5 | 52.4 | 93.4 |
| bottle | 2.6 | 69.9 | 4.4 | 81.5 | 39.0 | 81.7 |
| hazelnut | 15.3 | 81.3 | 22.4 | 93.6 | 48.9 | 94.5 |
| leather | 0.6 | 86.0 | 8.7 | 92.6 | 74.73 | 95.4 |
| cable | 11.0 | 39.4 | 14.8 | 64.6 | 45.6 | 64.0 |
| capsule | 12.2 | 63.8 | 16.8 | 88.4 | 18.0 | 87.8 |
| grid | 1.3 | 49.3 | 9.5 | 75.2 | 2.9 | 78.2 |
| pill | 7.9 | 66.9 | 11.5 | 90.0 | 33.6 | 90.3 |
| transistor | 3.0 | 45.5 | 9.5 | 57.7 | 20.2 | 60.3 |
| metal_nut | 3.8 | 39.7 | 5.5 | 71.8 | 42.3 | 70.3 |
| screw | 24.0 | 70.2 | 6.1 | 87.9 | 56.0 | 90.3 |
| toothbrush | 10.1 | 67.9 | 18.1 | 88.7 | 60.5 | 89.1 |
| zipper | 16.2 | 72.0 | 1.1 | 65.3 | 50.8 | 71.8 |
| tile | 12.3 | 54.5 | 9.5 | 87.8 | 10.4 | 84.9 |
| wood | 3.4 | 56.3 | 17.9 | 92.2 | 11.5 | 89.6 |
| Mean | 8.8 | 61.9 | 11.1 | 81.9 | 37.8 | 82.8 |

Table 13: Fine-grained comparison experiment of anomaly detection performance (AUROC) across data classes in MVTec AD.

| Object name | CLIP | WinCLIP | CoOp | AnomalyCLIP | AdaCLIP | GlocalCLIP |
|---|---|---|---|---|---|---|
| Carpet | 87.5 | 99.3 | 99.4 | 100 | 100 | 100 |
| Bottle | 97.9 | 98.6 | 93.3 | 88.7 | 96.8 | 89.4 |
| Hazelnut | 70.5 | 92.3 | 66.8 | 97.9 | 95.5 | 97.4 |
| Leather | 99.4 | 100 | 97.3 | 99.8 | 99.9 | 99.8 |
| Cable | 72.7 | 85.0 | 66.1 | 69.3 | 73.8 | 70.1 |
| Capsule | 75.9 | 68.7 | 75.5 | 87.8 | 86.2 | 89.4 |
| Grid | 95.6 | 99.2 | 97.9 | 97.8 | 99.2 | 97.8 |
| Pill | 64.0 | 81.5 | 76.5 | 81.3 | 88.2 | 80.5 |
| Transistor | 68.5 | 89.1 | 62.7 | 92.9 | 81.8 | 92.9 |
| Metal_nut | 78.2 | 96.2 | 65.9 | 92.5 | 80.5 | 89.3 |
| Screw | 84.2 | 71.7 | 88.5 | 84.4 | 80.1 | 86.2 |
| Toothbrush | 78.3 | 85.3 | 73.3 | 85 | 91.9 | 86.4 |
| Zipper | 80.9 | 91.2 | 78.5 | 97.6 | 95.8 | 98.3 |
| Tile | 96.2 | 99.9 | 95.6 | 100 | 99.9 | 100 |
| Wood | 99.0 | 97.6 | 95.1 | 97.1 | 98.3 | 97.4 |
| Mean | 83.3 | 90.4 | 82.1 | 91.5 | 91.2 | 91.7 |

Table 14: Fine-grained comparison experiment of anomaly detection performance (AP) across data classes in MVTec AD.

| Object name | CLIP | WinCLIP | CoOp | AnomalyCLIP | AdaCLIP | GlocalCLIP |
|---|---|---|---|---|---|---|
| Carpet | 96.4 | 99.8 | 99.8 | 100 | 100 | 100 |
| Bottle | 99.5 | 99.5 | 98.1 | 96.8 | 99.0 | 97 |
| Hazelnut | 84.7 | 96.0 | 82.2 | 99.0 | 97.3 | 98.7 |
| Leather | 99.8 | 100 | 99.1 | 99.9 | 100 | 99.9 |
| Cable | 83.9 | 89.8 | 77.8 | 80.6 | 84.3 | 81.3 |
| Capsule | 93.9 | 90.5 | 94.3 | 97.4 | 97.0 | 97.8 |
| Grid | 98.5 | 99.7 | 99.2 | 99.4 | 99.7 | 99.3 |
| Pill | 91.3 | 96.4 | 94.6 | 95.3 | 97.4 | 94.9 |
| Transistor | 66.1 | 84.9 | 63.2 | 90.9 | 82.9 | 90.8 |
| Metal_nut | 93.7 | 99.1 | 91.1 | 98.1 | 95.43 | 97.4 |
| Screw | 93.9 | 87.7 | 95.9 | 94.2 | 90.7 | 94.9 |
| Toothbrush | 90.9 | 94.5 | 84.6 | 93.4 | 97.1 | 94.5 |
| Zipper | 94.6 | 97.5 | 94.0 | 99.3 | 98.9 | 99.5 |
| Tile | 98.6 | 100 | 98.5 | 100 | 99.9 | 100 |
| Wood | 99.7 | 99.3 | 98.5 | 99.2 | 99.5 | 99.3 |
| Mean | 92.4 | 95.6 | 91.4 | 96.2 | 95.9 | 96.4 |

Table 15: Fine-grained comparison experiment of anomaly localization performance (AUROC) across data classes in VisA.

| Object name | CLIP | WinCLIP | CoOp | AnomalyCLIP | AdaCLIP | GlocalCLIP |
|---|---|---|---|---|---|---|
| Candle | 5.8 | 87.0 | 9.5 | 98.7 | 98.8 | 98.7 |
| Capsules | 33.7 | 79.9 | 26.5 | 95.0 | 98.3 | 95.6 |
| Cashew | 68.6 | 84.7 | 63.9 | 92.9 | 94.9 | 93.8 |
| Chewinggum | 8.8 | 95.4 | 17.5 | 99.1 | 99.6 | 99.2 |
| Fryum | 59.8 | 87.7 | 62.5 | 94 | 93.4 | 95.2 |
| Macaroni1 | 49.5 | 50.5 | 29.5 | 98.2 | 99.0 | 98.6 |
| Macaroni2 | 56.7 | 45.1 | 46.8 | 97.3 | 98.1 | 97.6 |
| Pcb1 | 57.7 | 38.7 | 39.1 | 93.7 | 92.15 | 95.9 |
| Pcb2 | 59.5 | 58.7 | 42.3 | 92.3 | 90.4 | 93.3 |
| Pcb3 | 61.8 | 75.9 | 63.8 | 88.1 | 88.9 | 88.4 |
| Pcb4 | 35.8 | 91.4 | 55.9 | 95.9 | 95.3 | 96.2 |
| Pipe_fryum | 76.7 | 83.7 | 48.2 | 98.3 | 97.3 | 98.8 |
| Mean | 47.9 | 73.2 | 42.1 | 95.3 | 95.5 | 95.9 |

Table 16: Fine-grained comparison experiment of anomaly localization performance (PRO) across data classes in VisA.

| Object name | CLIP | WinCLIP | CoOp | AnomalyCLIP | AdaCLIP | GlocalCLIP |
|---|---|---|---|---|---|---|
| Candle | 0.4 | 77.6 | 2.2 | 96.1 | 72.7 | 95.5 |
| Capsules | 13.1 | 39.4 | 3.5 | 79.4 | 90.7 | 82.1 |
| Cashew | 24.8 | 78.9 | 7.4 | 83.2 | 77.5 | 92.0 |
| Chewinggum | 8.2 | 68.7 | 6.4 | 90.2 | 60.2 | 89.5 |
| Fryum | 19.0 | 74.7 | 12.5 | 81.8 | 61.6 | 83.8 |
| Macaroni1 | 10.3 | 24.6 | 11.2 | 88.1 | 84.9 | 91.1 |
| Macaroni2 | 33.1 | 8.2 | 25.9 | 82.2 | 83.4 | 84.4 |
| Pcb1 | 17.5 | 21.0 | 3.1 | 82.0 | 73.0 | 88.0 |
| Pcb2 | 21.6 | 20.4 | 9.8 | 77.2 | 79.0 | 81.6 |
| Pcb3 | 20.0 | 44.3 | 27.1 | 76.4 | 76.2 | 75.3 |
| Pcb4 | 10.9 | 74.4 | 30.5 | 89.7 | 84.6 | 90.9 |
| Pipe_fryum | 14.7 | 80.4 | 6.5 | 95 | 89.7 | 96.3 |
| Mean | 16.1 | 51.1 | 12.2 | 85.1 | 77.8 | 87.5 |

Table 17: Fine-grained comparison experiment of anomaly detection performance (AUROC) across data classes in VisA.

| Object name | CLIP | WinCLIP | CoOp | AnomalyCLIP | AdaCLIP | GlocalCLIP |
|---|---|---|---|---|---|---|
| Candle | 92.9 | 95.0 | 91.0 | 78.4 | 92.4 | 72.6 |
| Capsules | 59.0 | 79.5 | 58.8 | 85.5 | 90.9 | 91.4 |
| Cashew | 69.0 | 91.2 | 90.4 | 71.2 | 82.5 | 88.7 |
| Chewinggum | 93.5 | 95.4 | 97.1 | 97.3 | 97.1 | 97.2 |
| Fryum | 77.9 | 73.9 | 86.4 | 89.0 | 91.7 | 91.9 |
| Macaroni1 | 68.0 | 79.3 | 72.9 | 88.2 | 72.6 | 86.1 |
| Macaroni2 | 66.7 | 67.0 | 69.5 | 74.4 | 50.6 | 78.9 |
| Pcb1 | 59.8 | 72.3 | 72.9 | 83.3 | 91.7 | 83.4 |
| Pcb2 | 48.6 | 46.9 | 65.3 | 61.5 | 65.8 | 62.8 |
| Pcb3 | 65.1 | 63.9 | 60.9 | 61.0 | 66.7 | 65.1 |
| Pcb4 | 74.1 | 74.2 | 74.0 | 94.0 | 87.1 | 94.5 |
| Pipe_fryum | 85.8 | 67.8 | 92.1 | 92.9 | 91.1 | 91.5 |
| Mean | 71.7 | 75.6 | 77.7 | 81.4 | 81.7 | 83.7 |

Table 18: Fine-grained comparison experiment of anomaly detection performance (AP) across data classes in VisA.

| Object name | CLIP | WinCLIP | CoOp | AnomalyCLIP | AdaCLIP | GlocalCLIP |
|---|---|---|---|---|---|---|
| Candle | 94.0 | 95.5 | 92.4 | 79.8 | 93.4 | 73.0 |
| Capsules | 73.3 | 87.9 | 69.8 | 90.9 | 93.7 | 94.6 |
| Cashew | 84.3 | 96.2 | 96.4 | 87.2 | 92.4 | 95.1 |
| Chewinggum | 97.1 | 98.1 | 98.7 | 98.9 | 98.8 | 98.8 |
| Fryum | 89.2 | 87.2 | 93.4 | 94.9 | 96.1 | 96.1 |
| Macaroni1 | 69.9 | 80.2 | 77.6 | 87.9 | 72.5 | 86.3 |
| Macaroni2 | 66.2 | 65.0 | 72.2 | 71.7 | 54.5 | 78.5 |
| Pcb1 | 57.4 | 73.4 | 76.2 | 83.3 | 90.1 | 83.4 |
| Pcb2 | 52.5 | 46.1 | 65.3 | 64.9 | 62.6 | 65.3 |
| Pcb3 | 66.4 | 63.3 | 62.5 | 68.6 | 70.3 | 72.0 |
| Pcb4 | 76.4 | 70.0 | 77.3 | 94.4 | 88.9 | 94.9 |
| Pipe_fryum | 92.9 | 82.1 | 96.6 | 96.5 | 95.1 | 95.8 |
| Mean | 76.6 | 78.8 | 81.5 | 84.9 | 84.0 | 86.2 |

Table 19: Fine-grained comparison experiment of anomaly localization performance (AUROC) across data classes in MPDD.

| Object name | CLIP | WinCLIP | CoOp | AnomalyCLIP | AdaCLIP | GlocalCLIP |
|---|---|---|---|---|---|---|
| Bracket_black | 42.8 | 46.4 | 21.8 | 95.6 | 95.2 | 95.9 |
| Bracket_brown | 51.0 | 56.2 | 43.6 | 94.4 | 93.9 | 95 |
| Bracket_white | 16.0 | 72.2 | 16.7 | 99.7 | 98.2 | 99.7 |
| Connector | 81.9 | 79.0 | 70.4 | 96.9 | 96.9 | 97.5 |
| Metal_plate | 28.3 | 95.7 | 25.7 | 92.8 | 95.0 | 93 |
| Tubes | 35.0 | 77.6 | 24.1 | 98.0 | 99.1 | 98.3 |
| Mean | 42.5 | 71.2 | 33.7 | 96.2 | 96.4 | 96.6 |

Table 20: Fine-grained comparison experiment of anomaly localization performance (PRO) across data classes in MPDD.

| Object name | CLIP | WinCLIP | CoOp | AnomalyCLIP | AdaCLIP | GlocalCLIP |
|---|---|---|---|---|---|---|
| Bracket_black | 12.9 | 13.6 | 1.3 | 83.9 | 51.3 | 87.4 |
| Bracket_brown | 32.5 | 12.4 | 29.8 | 77.5 | 48.6 | 78.4 |
| Bracket_white | 11.6 | 43.9 | 2.0 | 98.3 | 47.8 | 98.6 |
| Connector | 44.5 | 44.6 | 41.8 | 89.2 | 78.3 | 90.1 |
| Metal_plate | 6.6 | 83.7 | 3.2 | 84.1 | 52.4 | 85.6 |
| Tubes | 11.0 | 44.7 | 6.2 | 92.3 | 94.7 | 93.7 |
| Mean | 19.8 | 40.5 | 14.1 | 87.5 | 62.2 | 89.0 |

Table 21: Fine-grained comparison experiment of anomaly detection performance (AUROC) across data classes in MPDD.

| Object name | CLIP | WinCLIP | CoOp | AnomalyCLIP | AdaCLIP | GlocalCLIP |
|---|---|---|---|---|---|---|
| Bracket_black | 61.4 | 40.8 | 77.8 | 66.0 | 58.9 | 64.6 |
| Bracket_brown | 83.3 | 33.0 | 56.9 | 59.2 | 53.5 | 59.7 |
| Bracket_white | 64.9 | 41.7 | 68.3 | 64.6 | 59.8 | 68.6 |
| Connector | 75.7 | 79.3 | 77.9 | 89.3 | 73.3 | 89.3 |
| Metal_plate | 61.5 | 95.6 | 92.1 | 87.0 | 88.7 | 88.6 |
| Tubes | 80.7 | 78.7 | 83.2 | 95.4 | 98.5 | 94.6 |
| Mean | 71.2 | 61.5 | 76 | 76.9 | 72.1 | 77.6 |

Table 22: Fine-grained comparison experiment of anomaly detection performance (AP) across data classes in MPDD.

| Object name | CLIP | WinCLIP | CoOp | AnomalyCLIP | AdaCLIP | GlocalCLIP |
|---|---|---|---|---|---|---|
| Bracket_black | 68.6 | 56.5 | 79.6 | 71.6 | 64.7 | 70.8 |
| Bracket_brown | 91.4 | 59.8 | 72.6 | 77.9 | 71.5 | 78.3 |
| Bracket_white | 71.9 | 50.3 | 66.6 | 66.1 | 59.5 | 70.7 |
| Connector | 62.5 | 61.3 | 61.0 | 79.3 | 64.7 | 78.7 |
| Metal_plate | 84.0 | 98.3 | 97.0 | 95.2 | 96.1 | 95.7 |
| Tubes | 90.8 | 89.1 | 92.8 | 98.0 | 99.4 | 97.7 |
| Mean | 78.2 | 69.2 | 78.3 | 81.4 | 76.0 | 82.0 |

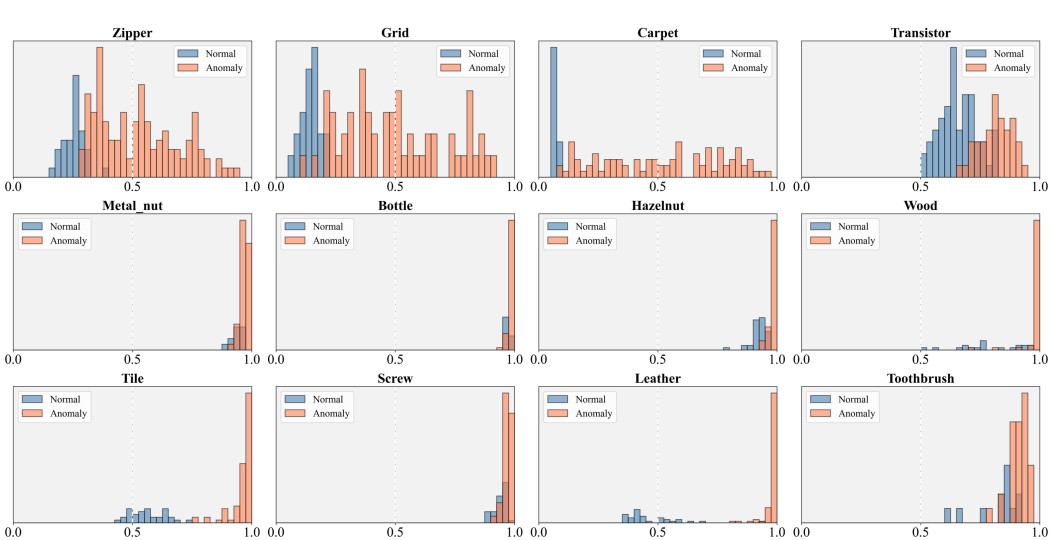

Figure 9: Visualization of histograms illustrating cosine similarity measurements for each class within the MVTec AD dataset.

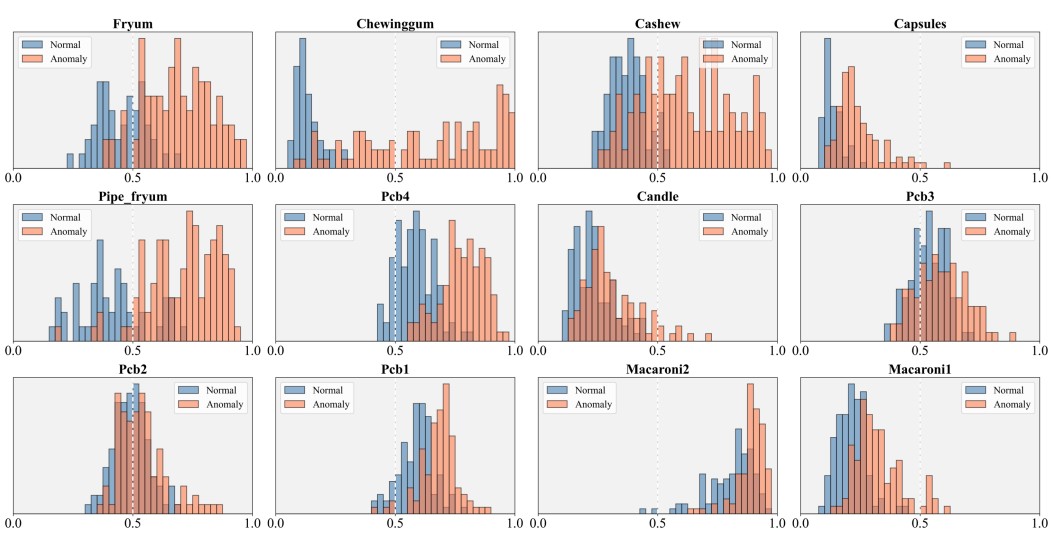

Figure 10: Visualization of histograms illustrating cosine similarity measurements for each class within the VisA dataset.

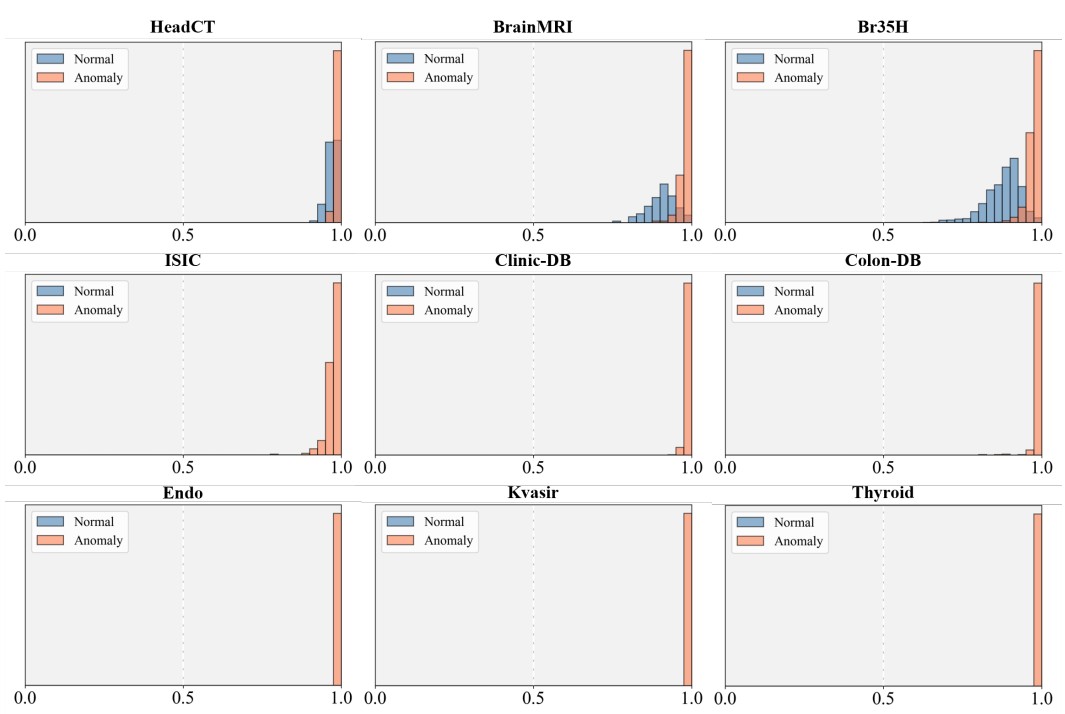

Figure 11: Visualization of histograms illustrating cosine similarity measurements for each class within the medical domain datasets.

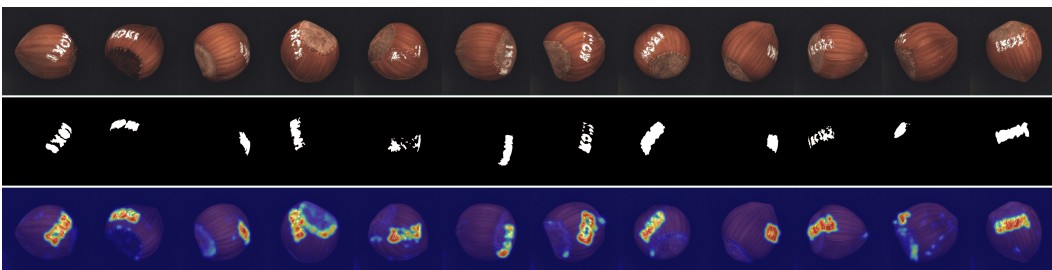

Figure 12: Anomaly localization maps for the hazelnut class in MVTec AD. The first row represents the input, the second row shows the ground truth (GT), and the last row illustrates the localization results from GlocalCLIP.

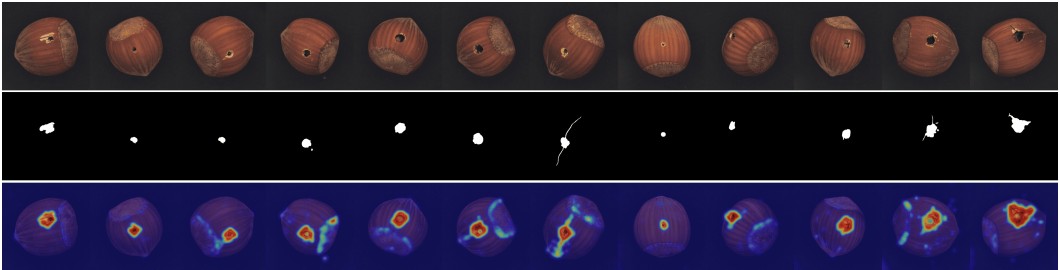

Figure 13: Anomaly localization maps for the hazelnut class in MVTec AD. The first row represents the input, the second row shows the ground truth (GT), and the last row illustrates the localization results from GlocalCLIP.

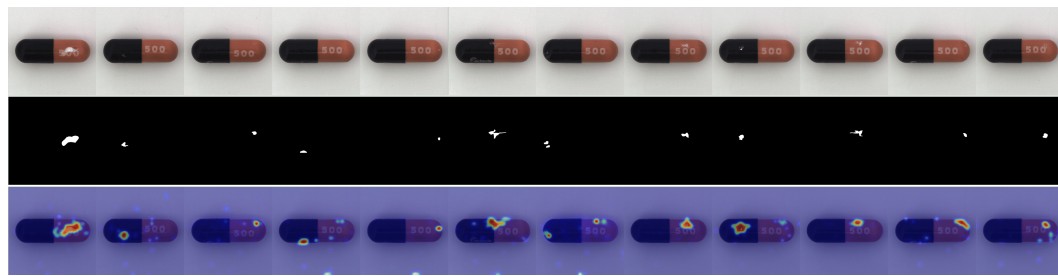

Figure 14: Anomaly localization maps for the capsule class in MVTec AD. The first row represents the input, the second row shows the ground truth (GT), and the last row illustrates the localization results from GlocalCLIP.

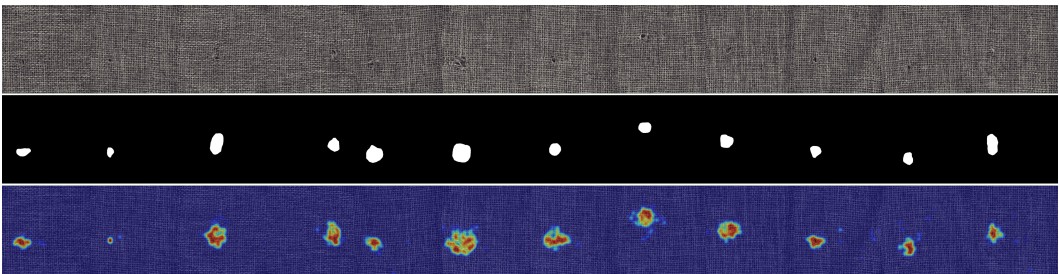

Figure 15: Anomaly localization maps for the carpet class in MVTec AD. The first row represents the input, the second row shows the ground truth (GT), and the last row illustrates the localization results from GlocalCLIP.

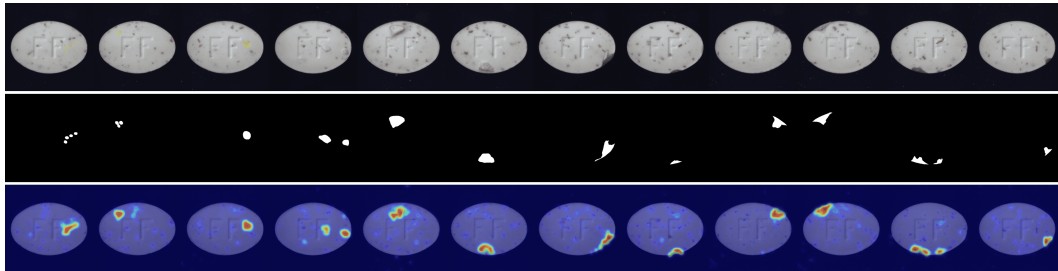

Figure 16: Anomaly localization maps for the pill class in MVTec AD. The first row represents the input, the second row shows the ground truth (GT), and the last row illustrates the localization results from GlocalCLIP.

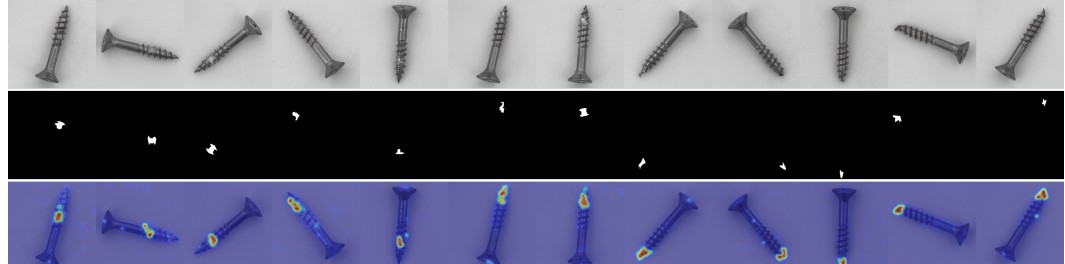

Figure 17: Anomaly localization maps for the screw class in MVTec AD. The first row represents the input, the second row shows the ground truth (GT), and the last row illustrates the localization results from GlocalCLIP.

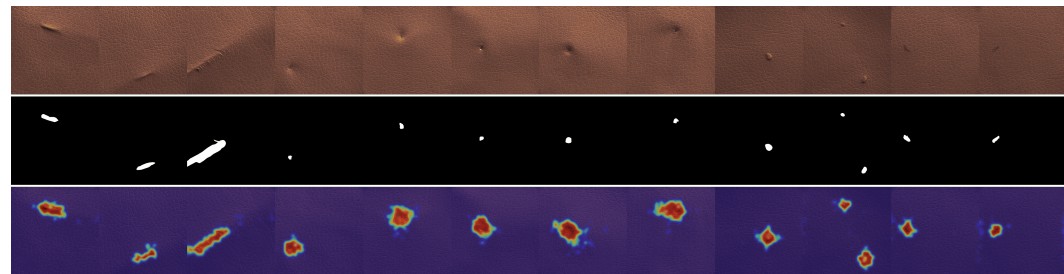

Figure 18: Anomaly localization maps for the leather class in MVTec AD. The first row represents the input, the second row shows the ground truth (GT), and the last row illustrates the localization results from GlocalCLIP.

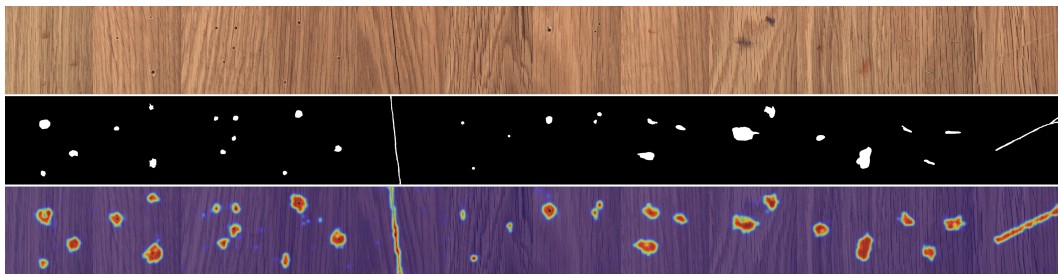

Figure 19: Anomaly localization maps for the wood class in MVTec AD. The first row represents the input, the second row shows the ground truth (GT), and the last row illustrates the localization results from GlocalCLIP.

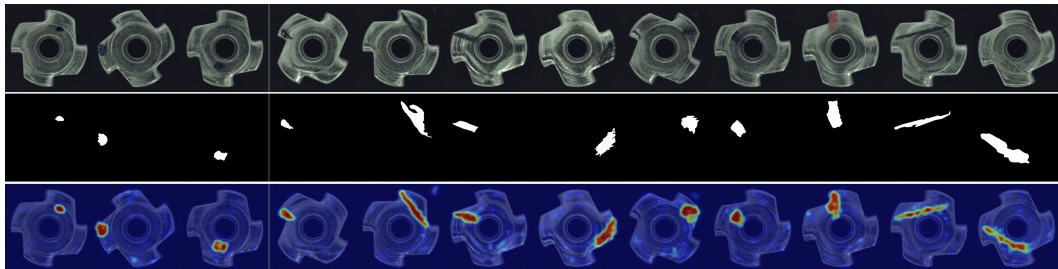

Figure 20: Anomaly localization maps for the metal nut class in MVTec AD. The first row represents the input, the second row shows the ground truth (GT), and the last row illustrates the localization results from GlocalCLIP.

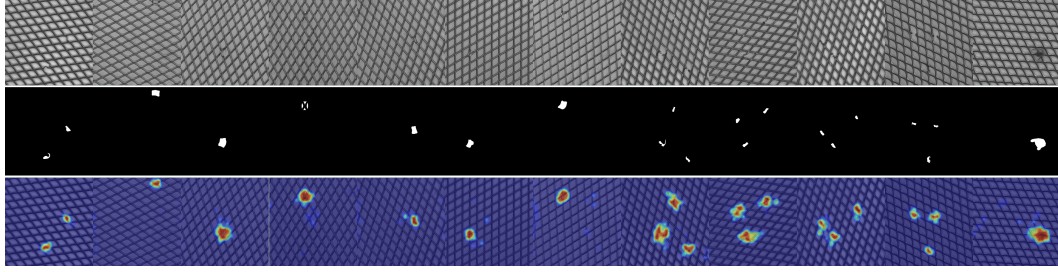

Figure 21: Anomaly localization maps for the grid class in MVTec AD. The first row represents the input, the second row shows the ground truth (GT), and the last row illustrates the localization results from GlocalCLIP.

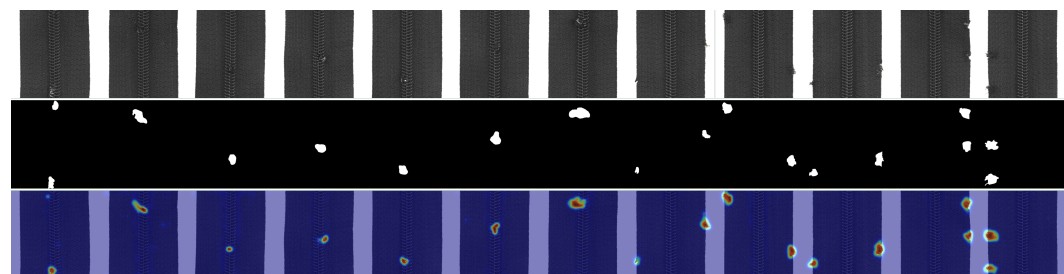

Figure 22: Anomaly localization maps for the zipper class in MVTec AD. The first row represents the input, the second row shows the ground truth (GT), and the last row illustrates the localization results from GlocalCLIP.

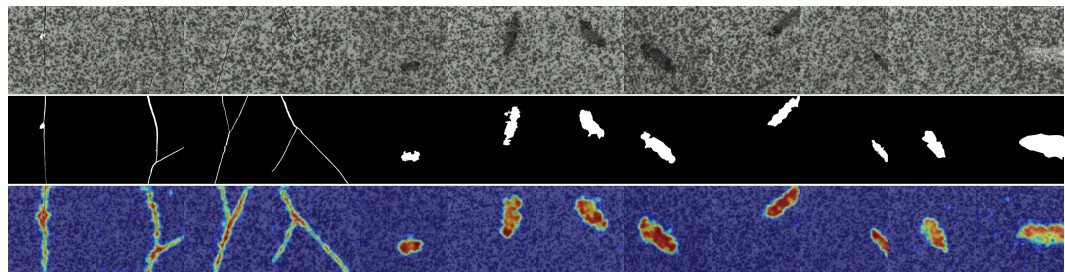

Figure 23: Anomaly localization maps for the tile class in MVTec AD. The first row represents the input, the second row shows the ground truth (GT), and the last row illustrates the localization results from GlocalCLIP.

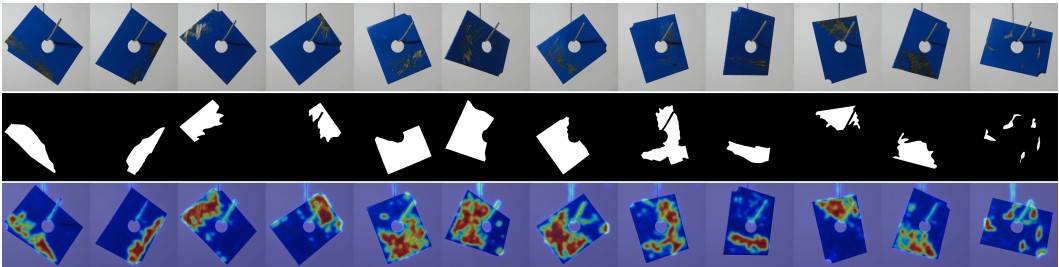

Figure 24: Anomaly localization maps for the metal plate class in MPDD. The first row represents the input, the second row shows the ground truth (GT), and the last row illustrates the localization results from GlocalCLIP.

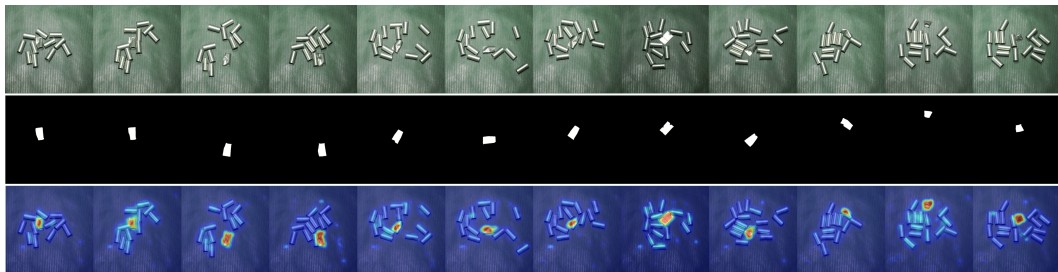

Figure 25: Anomaly localization maps for the tubes class in MPDD. The first row represents the input, the second row shows the ground truth (GT), and the last row illustrates the localization results from GlocalCLIP.

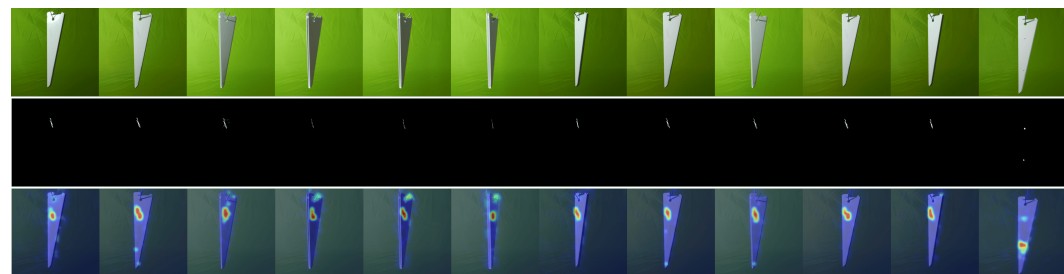

Figure 26: Anomaly localization maps for the white bracket class in MPDD. The first row represents the input, the second row shows the ground truth (GT), and the last row illustrates the localization results from GlocalCLIP.

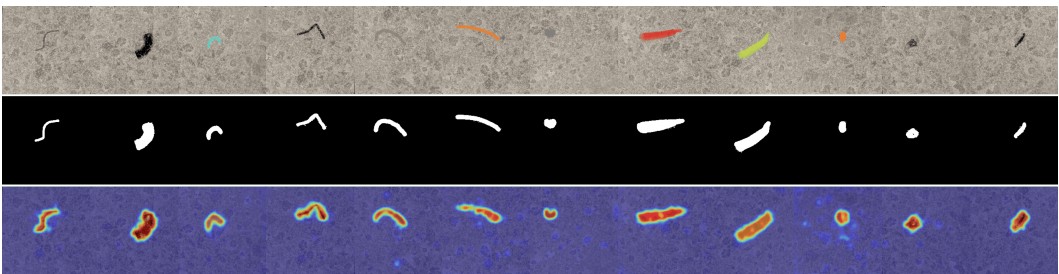

Figure 27: Anomaly localization maps for the blotchy class in DTD-Synthetic. The first row represents the input, the second row shows the ground truth (GT), and the last row illustrates the localization results from GlocalCLIP.

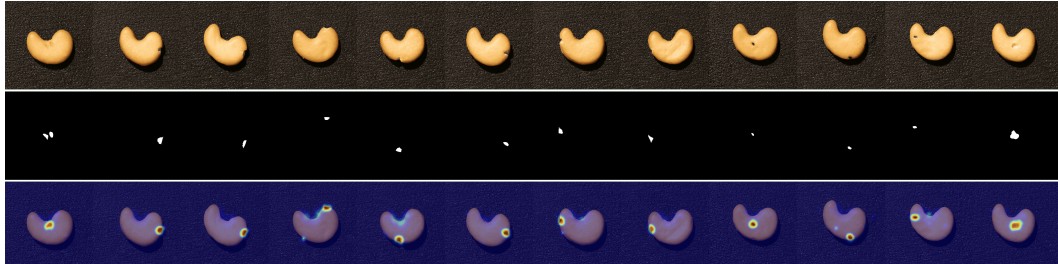

Figure 28: Anomaly localization maps for the cashew class in VisA. The first row represents the input, the second row shows the ground truth (GT), and the last row illustrates the localization results from GlocalCLIP.

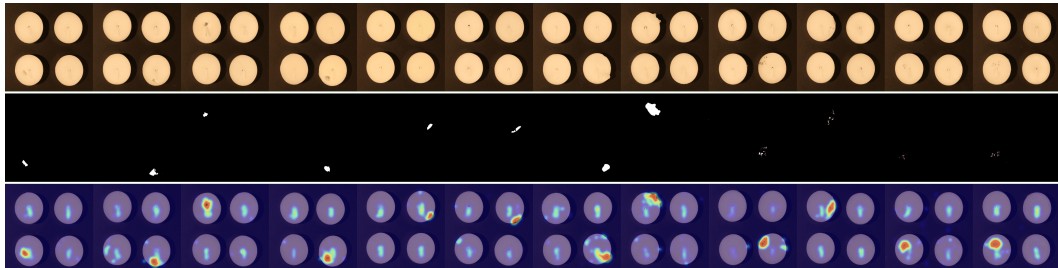

Figure 29: Anomaly localization maps for the candle class in VisA. The first row represents the input, the second row shows the ground truth (GT), and the last row illustrates the localization results from GlocalCLIP.

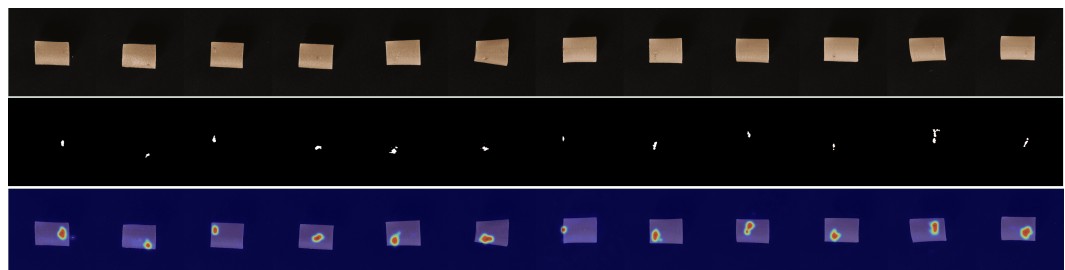

Figure 30: Anomaly localization maps for the pipe fryum class in VisA. The first row represents the input, the second row shows the ground truth (GT), and the last row illustrates the localization results from GlocalCLIP.

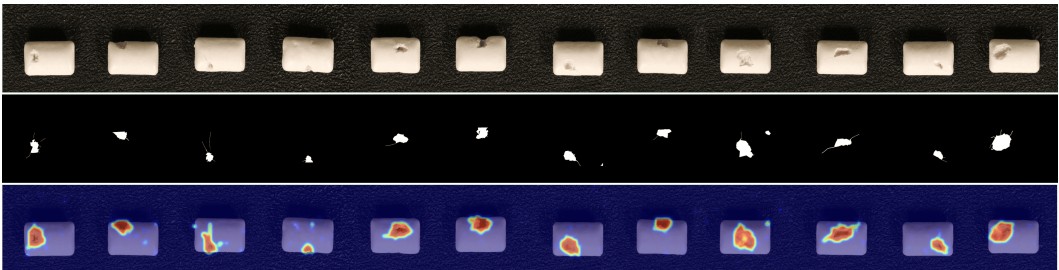

Figure 31: Anomaly localization maps for the chewinggum class in VisA. The first row represents the input, the second row shows the ground truth (GT), and the last row illustrates the localization results from GlocalCLIP.

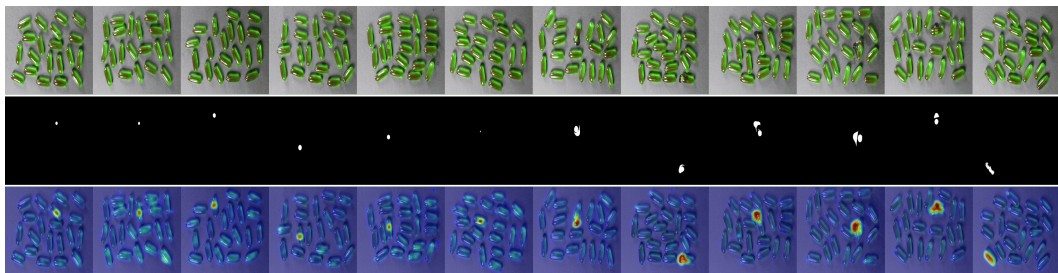

Figure 32: Anomaly localization maps for the capsules fryum class in VisA. The first row represents the input, the second row shows the ground truth (GT), and the last row illustrates the localization results from GlocalCLIP.

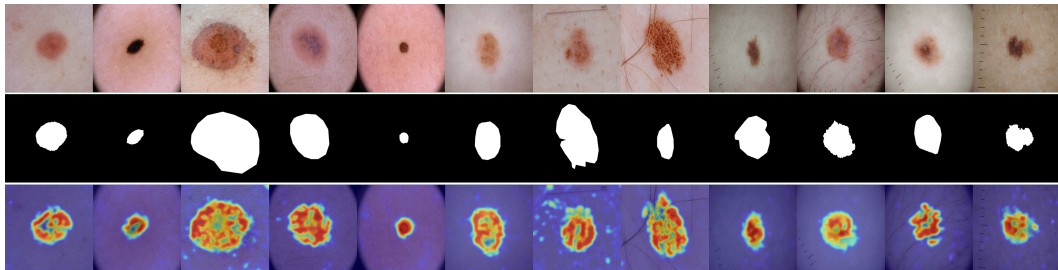

Figure 33: Anomaly localization maps for the skin class in ISIC. The first row represents the input, the second row shows the ground truth (GT), and the last row illustrates the localization results from GlocalCLIP.

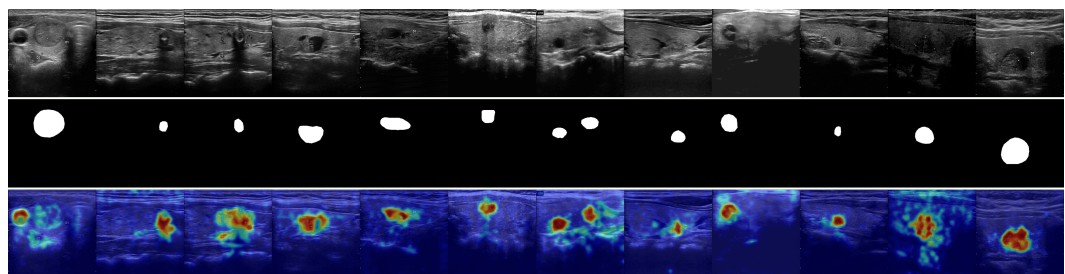

Figure 34: Anomaly localization maps for the thyroid class in Tn3K. The first row represents the input, the second row shows the ground truth (GT), and the last row illustrates the localization results from GlocalCLIP.

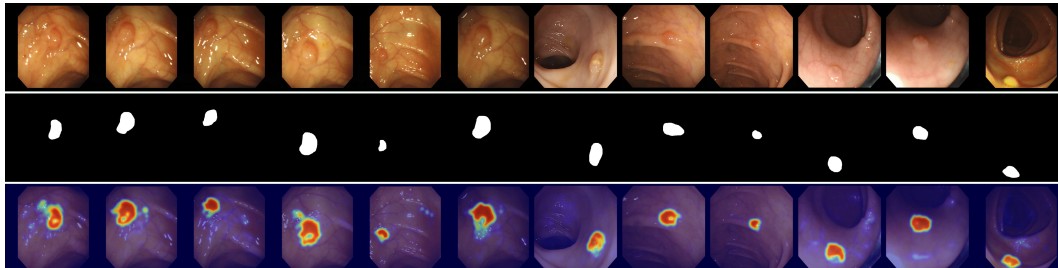

Figure 35: Anomaly localization maps for the colon class in CVC-ColonDB. The first row represents the input, the second row shows the ground truth (GT), and the last row illustrates the localization results from GlocalCLIP.

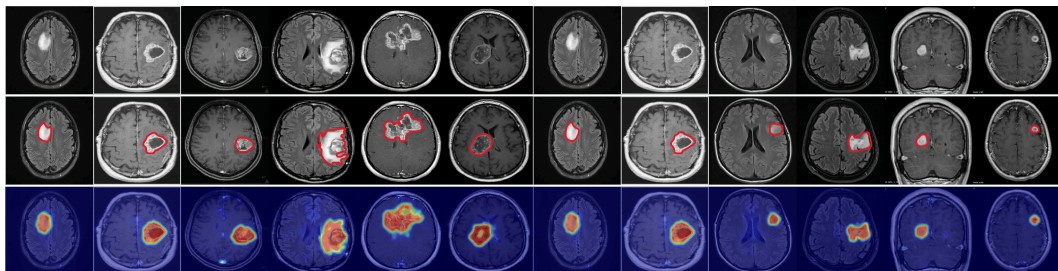

Figure 36: Anomaly localization maps for the brain class in BrainMRI. The first row represents the input, the second row shows the ground truth (GT), and the last row illustrates the localization results from GlocalCLIP.

