# OpenReview forum: "GlocalCLIP: Object-agnostic Global-Local Prompt Learning for Zero-shot Anomaly Detection"
_ICLR.cc/2025/Conference — Submitted to ICLR 2025_

### Official Review · Reviewer_bsxX · 2024-10-29

**Soundness:** 2
**Presentation:** 2
**Contribution:** 2
**Rating:** 5
**Confidence:** 4

**Summary:**

This paper aims to adapt CLIP for the challenging task of zero-shot anomaly detection. The authors propose enabling object-agnostic learning to capture both global and local anomaly semantics. Experimental results seem to demonstrate the effectiveness of the proposed method.

**Strengths:**

1. This paper focuses on a challenging and valuable field that is practical to the real world.
2. The authors conduct experiments across diverse datasets to support their claims.

**Weaknesses:**

1. This paper presents similar technological contributions and organization to AnomalyCLIP. However, the authors do not provide a comprehensive comparison and discussion with AnomalyCLIP. A detailed analysis in the introduction section is necessary to explicitly tell the main technological differences from AnomalyCLIP.
2. The manuscript lacks proper citations in several sections, notably in 3.2 (Prompt Design) and 3.5 (Training and Inference). The authors should carefully review and appropriately cite previous research to acknowledge foundational work in this field.
3. The illustration in the paper is unclear. For example, it is not evident what the term "anchor" refers to in Eq. 5, and the rationale for using V-V attention instead of other attention mechanisms, such as Q-Q or K-K attention, should be clearly explained.

**Questions:**

See Weaknesses

---

> ### Author Response · Authors · 2024-11-28
>
> > ### Q1. Difference between AnomalyCLIP and Ours.
>
> Thank you for your question regarding the differences between AnomalyCLIP [4] and our approach. As described in General response 1, we acknowledge and appreciate the foundational contributions of AnomalyCLIP and related works.
> To summarize, our method, GlocalCLIP, extends AnomalyCLIP by explicitly separating global and local information and employing Glocal Contrastive Learning. This design allows for complementary learning of both global and local perspectives, which plays a critical role in improving decision-making for anomaly detection tasks. The effectiveness of our approach is demonstrated through the results presented in Fig. 3 and Tables 1 and 2.
>
> We hope this addresses your concern, and we are happy to provide further clarification if needed.

---

> ### Author Response · Authors · 2024-11-28
>
> > ### Q2. Lack of Paper Citations in Key Sections.
>
> Thank you for reading thoroughly and helping us address citations.

---

> ### Author Response · Authors · 2024-11-28
>
> > ### Q3. Clarification of Illustrations and Rationale for Attention Mechanisms.
>
> Thank you for your question regarding anchor prompts. As explained in General response 3, the global prompt was selected as the anchor based on the hierarchical characteristics of image semantics. This choice ensures better alignment of local text embeddings relative to global normality and abnormality.
> For a detailed explanation and supporting results, please refer to General response 3 and the ablation study summarized in the table. If further clarification is needed, we would be happy to provide additional details.
>
> V-V attention proves to be advantageous due to its ability to preserve local visual semantics while minimizing bias and disturbance in the attention computation process. Unlike Q-Q and K-K attention, which rely on the original attention map formed by Q and K, V-V attention operates independently of the original attention map. We conducted an ablation study comparing Q-Q, K-K, and V-V attention. The results demonstrated that V-V attention effectively extracts local visual features as intended.
>
> | Attention | Industrial domain |  | Medical domain |  |
> |:---:|:---:|:---:|:---:|:---:|
> |  | **Pixel-level** | **Image-level** | **Pixel-level** | **Image-level** |
> | **Q-Q** | (74.3, 46.6) | (50.1, 73.2) | (79.6, 50.2) | (41.5, 49.0) |
> | **K-K** | (78.5, 52.6) | (50.8, 61.7) | (78.8, 46.5) | (68.6, 71.8)   |
> | **V-V**| (**95.3, 84.0**) | (**87.2, 89.8**) | (**90.3, 75.1**) | (**94.9, 95.4**) |

---

> > ### Comment · Reviewer_bsxX · 2024-11-29
> >
> > Thank you for your efforts in addressing my concerns. While my questions have largely been addressed, I still have reservations regarding whether the paper meets the high publication standards of ICLR, particularly in terms of its contribution to the field and its potential to inspire future research. As a result, I adjust my rating to borderline reject.

---

> ### Author Response · Authors · 2024-12-01
>
> Dear reviewer bsxX,
>
> We appreciate your thoughtful feedback and are glad to have addressed your concerns. We firmly believe that our work can contribute meaningfully to this field, and we are committed to further advancements in our future research.
>
> Best regards,
>
> Authors

---

### Official Review · Reviewer_1Cz8 · 2024-10-30

**Soundness:** 3
**Presentation:** 3
**Contribution:** 1
**Rating:** 3
**Confidence:** 5

**Summary:**

This work investigates the task of zero-shot anomaly detection and proposes a model named GlocalCLIP, which employs a dual-branch approach for both global and local modeling of image and text inputs based on CLIP. Results across multiple datasets demonstrate the effectiveness of the proposed method.

**Strengths:**

- The paper is clearly presented, including both text and images, making it easy to reproduce based on existing work.
- Quantitative and qualitative experiments on multiple datasets.

**Weaknesses:**

- The novelty of the method is limited compared to AnomalyCLIP. The framework still follows the standard Winclip framework and subsequent works. The authors' claim of the "first framework" is exaggerated, as prompt tuning and V-V attention have already been used in anomaly detection, particularly in AnomalyCLIP.
- The design of global and local branches is not novel. The authors should provide a more detailed explanation of what the two types of tokens model in zero-shot anomaly detection.
- The global loss does not use multi-stage feature fusion, unlike the local loss. Additionally, the novelty of the GCL loss is trivial, lacking any impressive design.
- The introduction resembles related work, and the motivation is not convincing.
- For Fig. 1, it is unclear whether the authors aim to highlight the novelty of the zero-shot anomaly detection task or the disadvantages of other settings. I disagree with the authors' claim in Sec. 2.4 that few-shot and multi-class models (e.g., UniAD) are unfriendly for practical applications.
- The improvement over AnomalyCLIP is not significant.
- Please report the differences in training overhead, computational costs, and efficiency comparisons of different methods.

**Questions:**

Please refer to the Weaknesses section.

---

> ### Author Response · Authors · 2024-11-28
>
> > ### Q1. Difference between AnomalyCLIP and Ours.
>
> Thank you for your question regarding the differences between AnomalyCLIP [4] and our approach. As described in General response 1, we acknowledge and appreciate the foundational contributions of AnomalyCLIP and related works.
> To summarize, our method, GlocalCLIP, extends AnomalyCLIP by explicitly separating global and local information and employing Glocal Contrastive Learning. This design allows for complementary learning of both global and local perspectives, which plays a critical role in improving decision-making for anomaly detection tasks. The effectiveness of our approach is demonstrated through the results presented in Fig. 3 and Tables 1 and 2.

---

> ### Author Response · Authors · 2024-11-28
>
> > ### Q2. Global and Local Branch Design.
>
> Thank you for your question regarding global and local prompts. As explained in General response 2, the third row of Fig. 4 demonstrates the differences between global and local prompts. Global prompts focus on capturing the overall semantics of an object, while local prompts concentrate on detecting fine-grained, localized defects. This distinction is further supported by the visualizations in Appendix Fig. 6 to Fig. 8, which highlight their complementary roles in representing normality and abnormality.
>
> We hope this provides sufficient clarification. Please let us know if further details are required.

---

> ### Author Response · Authors · 2024-11-28
>
> > ### Q3. Concerns Regarding Global Loss and GCL Loss Design.
>
> The reason for not utilizing multi-stage outputs in the global loss is that the final layer's output best captures the class semantics of the entire image. In contrast, for the local loss, multi-stage outputs were used because the reception field of the ViT changes depending on the depth of the blocks. This behavior is supported by Appendix Fig. 11 in the ViT [6]. Therefore, the local loss leverages multi-stage outputs to account for these variations.
>
> Although GCL adopts a simple structure, we believe its use is well-justified. The primary goal of GCL is to maximize the interaction between Global and Local Prompts. Its effectiveness is demonstrated in Table 3 and Figure 4. While it may appear simple, it is an optimally designed loss structure that aligns with its intended purpose.
>
> [Reference]
>
> [6] Dosovitskiy, A. (2020). An image is worth 16x16 words: Transformers for image recognition at scale. arXiv preprint arXiv:2010.11929.

---

> ### Author Response · Authors · 2024-11-28
>
> > ### Q4,5,6. Weak Introduction, Lack of Convincing Motivation, Clarification of Fig. 1.
>
> We have revised the Introduction, Related Work, and Motivation sections. After further reflection on Fig. 1, we concluded that illustrating the comparison of prompt design would better highlight the contributions of our work. In the revised Introduction, we have acknowledged prior studies more comprehensively. Additionally, the Motivation section now emphasizes the difference between global and local representations in CLIP's results, which served as the foundation for our research approach.
>
> Thank you sincerely for helping us refine and improve our paper.

---

> ### Author Response · Authors · 2024-11-28
>
> > ### Q7. Marginal Improvement Over AnomalyCLIP.
>
> AnomalyCLIP demonstrates SOTA performance in ZSAD. As shown in Tables 1 and 2, as well as the Appendix, the improvements across all 15 datasets are substantial and can be considered significant.

---

> ### Author Response · Authors · 2024-11-28
>
> > ### Q8. Efficiency, Computational Costs, and Training Overhead.
>
> We kindly note that training efficiency is not the primary focus of this study, and we appreciate your understanding.

---

> ### Comment · Reviewer_1Cz8 · 2024-11-28
>
> My concerns have not been adequately addressed. The author should provide more explicit qualitative and quantitative comparative analyses. I have also reviewed comments from other reviewers, Reviewers #5eoh and #bsxX, who share similar concerns. However, the author has not directly provided specific comparisons. Additionally, the author should highlight the revised sections in the revision. I keep my original rating.

---

> ### Author Response · Authors · 2024-11-28
>
> We regret that your concerns were not fully addressed. We have revised the manuscript by incorporating the feedback and questions from other reviewers to further enhance the contributions and additional details. As you suggested, the revised submission will clearly highlight the updated sections for your review. Additionally, detailed responses to each comment have been provided in the corresponding sections, and we kindly ask for your confirmation on those. Regarding your feedback on the insufficiency of the quantitative and qualitative results, it is still unclear which specific aspects need improvement. If you could provide further clarification on the areas that fall short, it would greatly help us refine this study and contribute to the development of our future research efforts.

---

### Official Review · Reviewer_5eoh · 2024-11-01

**Soundness:** 2
**Presentation:** 3
**Contribution:** 2
**Rating:** 5
**Confidence:** 5

**Summary:**

The authors propose GlocalCLIP, which introduces separable global and local prompts through an object-agnostic glocal semantic prompt design and jointly optimizes them to address the zero-shot anomaly detection task. They also employ contrastive learning to enhance the learning of global and local visual features. The effectiveness of the method is validated on 15 datasets.

**Strengths:**

The authors propose GlocalCLIP, which introduces separable global and local prompts through an object-agnostic glocal semantic prompt design and jointly optimizes them to address the zero-shot anomaly detection task. They also employ contrastive learning to enhance the learning of global and local visual features.

**Weaknesses:**

See questions.

**Questions:**

1. The structure of the proposed method appears to be almost identical to AnomalyCLIP, except for improvements in prompt design and contrastive learning. There is no fundamental change at the framework level.
2. The design of the V-V attention layer has already been experimented with and used in AnomalyCLIP.
3. I believe the incremental experiments in Table 3 should be conducted starting from AnomalyCLIP to demonstrate the effectiveness of the newly proposed innovations.
4. In Table 3, adding F4, which corresponds to GCL in Figure 4, does not seem to significantly improve the pixel-level metrics and even shows some decline. However, in Figure 4, adding GCL shows a clear difference in pixel-level performance compared to not adding GCL. Please explain the reason for this discrepancy.

---

> ### Author Response · Authors · 2024-11-28
>
> > ### Q1. Difference between AnomalyCLIP and Ours.
>
> Thank you for your question regarding the differences between AnomalyCLIP and our approach. As described in General response 1, we acknowledge and appreciate the foundational contributions of AnomalyCLIP and related works.
> To summarize, our method, GlocalCLIP, extends AnomalyCLIP by explicitly separating global and local information and employing Glocal Contrastive Learning. This design allows for complementary learning of both global and local perspectives, which plays a critical role in improving decision-making for anomaly detection tasks. The effectiveness of our approach is demonstrated through the results presented in Fig. 3 and Tables 1 and 2.
>
> We hope this addresses your concern, and we are happy to provide further clarification if needed.

---

> ### Author Response · Authors · 2024-11-28
>
> > ### Q2. Validation of V-V Attention Layer Usage.
>
> Recent studies on anomaly detection using CLIP, such as AnomalyCLIP, PromptAD, and AnoVL [5], have demonstrated the effectiveness of V-V attention. To validate this, we conducted various experiments from an image perspective. When using V-V attention, methods such as introducing a linear layer for post-adaptation or employing image-to-text cross-attention proved to be less suitable. Instead, the use of QKV attention yielded better results.
> Thus, we concluded that V-V attention provides a simpler and more efficient approach to learning anomalous patterns, which is why we decided to retain the overall structure in our research. To further optimize the effectiveness of this structure, we focused on developing prompt learning methodologies. The ablation study on attention is as follows:
>
> | Attention | Industrial domain |  | Medical domain |  |
> |:---:|:---:|:---:|:---:|:---:|
> |  | **Pixel-level** | **Image-level** | **Pixel-level** | **Image-level** |
> | **Q-Q** | (74.3, 46.6) | (50.1, 73.2) | (79.6, 50.2) | (41.5, 49.0) |
> | **K-K** | (78.5, 52.6) | (50.8, 61.7) | (78.8, 46.5) | (68.6, 71.8)   |
> | **V-V**| (**95.3, 84.0**) | (**87.2, 89.8**) | (**90.3, 75.1**) | (**94.9, 95.4**) |
>
> [Reference]
>
> [5] Deng, H., Zhang, Z., Bao, J., & Li, X. (2023). Anovl: Adapting vision-language models for unified zero-shot anomaly localization. arXiv preprint arXiv:2308.15939.

---

> ### Author Response · Authors · 2024-11-28
>
> > ### Q3. Incremental Experiments Starting from AnomalyCLIP.
>
> While we acknowledge prior research, we also aimed to demonstrate the necessity of reproduction in our study. In Table 3, F2 is identical to AnomalyCLIP except for the semantic prompt design, allowing for a direct comparison with Tables 1 and 2. Through F2, we aimed to highlight the effectiveness of semantic prompt design with deep-text prompts.
> Next, F3 demonstrates the benefits of separating global and local prompts. The case where global and local prompts are separated but semantic design is not applied can be seen in Table 4. Finally, F4 showcases the effectiveness of GCL. We would also like to share an ablation table aligned with your suggestions to further clarify our findings.
>
> | Module | Industrial domain |  | Medical domain |  |
> |:---:|:---:|:---:|:---:|:---:|
> |  | **Pixel-level** | **Image-level** | **Pixel-level** | **Image-level** |
> | **AnomalyCLIP** | (94.2, 81.0) | (86.1, 88.7) | (89.1, 72.0) | (94.4, 95.0) |
> | **+Semantic design** | (95.0, 82.4) | (85.6, 88.3) | (90.0, 74.5) | (94.6, 95.2)   |
> | **+Global local branch**| (95.3, **84.0**) | (86.2, 88.5) |(90.2, 74.4) | (89.8, 91.1) |
> | **+GCL** | (**95.3**, 83.3) | (**86.7, 89.3**) | (**90.3, 74.8**) | (**94.9, 95.4**) |
>
> We noted improvements in the Industrial domain, with Pixel-level AUROC increasing by 1.2%, AUPRO by 3.7%, Image-level AUROC by 0.7%, and AP by 1.4%. Similarly, in the Medical domain, Pixel-level AUROC increased by 1.3%, AUPRO by 3.9%, Image-level AUROC by 0.5%, and AP by 0.4%.
>
> Please let us know if further details are required. Thank you for your valuable feedback.

---

> ### Author Response · Authors · 2024-11-28
>
> > ### Q4. Discrepancy Between F4 in Table 3 and GCL in Figure 4.
>
> In Table 3, the effectiveness of GCL (F4) was influenced by the characteristics of the dataset. GCL primarily focuses on learning complementary information between global and local prompts. However, cases where GCL was less effective can be explained using the MVTecAD and VisA datasets.
> The key difference between these two datasets lies in the composition of objects within the images. MVTecAD mainly consists of single objects, with a focus on detecting potential defects within those objects. In contrast, VisA often includes multiple instances of objects within the same class (e.g., candles, capsules, macaroni) or overlapping defects across objects (e.g., cashew, fryum, pipefryum). Such multi-object scenarios increase the complexity of anomaly detection as defects may be confined to only one of the multiple objects.
> In images containing multiple objects, referring to global information sometimes introduced confusion to local information, making it more effective to focus solely on local information. This might explain the performance decline observed in certain datasets when averaged.
> In summary, for datasets with multiple objects, independently analyzing global and local information proved to be more effective, while combining both types of information created better synergy for single-object scenarios. Nonetheless, the separation of global and local prompts consistently outperformed existing methods, and the significance of GCL varied depending on the characteristics of the dataset.

---

> ### Comment · Reviewer_5eoh · 2024-11-28
>
> I believe my concerns have not been adequately addressed. Firstly, the improvements based on AnomalyClip are limited. Secondly, the proposed Global Local Branch seems to offer limited enhancement in the medical field and even shows a decline at the image level. Lastly, while the qualitative experimental results indicate significant improvement in anomaly localization with GCL, this is not evident in the quantitative experiments. Additionally, I do not fully agree with the author's explanation regarding single and multiple objects in response to Q4. I suggest the author use blue color to mark changes in the revised version for easier review and submit the rebuttal promptly rather than after an extension deadline. Therefore, I decide to maintain the original score.

---

> ### Author Response · Authors · 2024-11-28
>
> We regret that your concerns were not fully addressed. Regarding AnomalyCLIP, We understand that it may seem limited in scope. However, We believe that the proposed prompt learning method demonstrates strong performance within the AnomalyCLIP framework and has the potential to achieve similarly positive results in other proposed methodological structures, showcasing its scalability and effectiveness. Secondly, regarding the performance reduction of F3, please refer to the response to reviewer fHMB's question. Additionally, the performance reduction of the global local branch in the medical domain is due to the fact that simply dividing branches can lead to a loss of complementary information. This is precisely why we proposed GCL to address this issue, and We would greatly appreciate it if you could take this into account. Lastly, regarding Q4, it would be incredibly helpful for us if you could provide more details about the reasons for your disagreement, as this would allow us to make more substantial improvements. For your convenience, we have highlighted the changes in the revised version of the paper in blue.
>
> Thank you once again for your detailed review and for taking the time to engage with our work. Your feedback has been invaluable in helping us improve our research, and we truly appreciate your interest and thoughtful input.

---

### Official Review · Reviewer_fHMB · 2024-11-04

**Soundness:** 3
**Presentation:** 3
**Contribution:** 2
**Rating:** 5
**Confidence:** 4

**Summary:**

This paper introduces a novel method for zero-shot anomaly detection. GlocalCLIP separates global and local prompts and optimize them together, enabling detection of general anomalies without object dependence. Through refined text prompts and a V-V attention layer for detailed image features, GlocalCLIP effectively captures both normal and abnormal patterns. Tested on 15 industrial and medical datasets, it outperforms current ZSAD methods.

**Strengths:**

+  An object-agnostic global  and local prompts are supposed to learn normal and abnormal patterns.
+ Align visual features and text features by jointly optimizing global and local prompts through contrastive learning to enhance robustness.
+  The proposed method demonstrate excellent performance on multiple datasets.

**Weaknesses:**

This paper primarily focuses on improvements to AnomalyCLIP, particularly with regard to enhancements in text prompts. The paper has some unclear explanations that can make it difficult to understand.

**Questions:**

+ How to demonstrate the differences between global prompts $g_n$, $g_a$and local prompts $l_n$, $l_a$?
+ Does 'semantic prompt design' refer to Sec3.3 ? Why does it perform poorly on 'single', and even lead to a decrease in performance in table 4?
+ What causes the F3 to reduce image-level performance so drastically in medical applications in table 3? Could you provide a detailed explanation and analysis?
+ What' s the anchor prompts?
+ Some minor issues, such as line 316.

---

> ### Author Response · Authors · 2024-11-28
>
> > ### Q1. Dfference between global prompts and local prompts.
>
> Thank you for your question regarding global and local prompts. As explained in General response 2, the third row of Fig. 4 demonstrates the differences between global and local prompts. Global prompts focus on capturing the overall semantics of an object, while local prompts concentrate on detecting fine-grained, localized defects. This distinction is further supported by the visualizations in Appendix Fig. 6 to Fig. 8, which highlight their complementary roles in representing normality and abnormality.

---

> ### Author Response · Authors · 2024-11-28
>
> > ### Q2. Semantic Prompt Design and Its Impact on Single-Task Performance.
>
> Thank you for your question regarding Table 4. While AUROC generally increased or remained stable, the decrease in AP within the Industrial domain can be interpreted as a reduced tendency to classify instances as anomalies. This may be due to the addition of normal learnable tokens to the prompts, which could introduce confusion when a single prompt is tasked with determining anomaly status. However, this issue appears to have been addressed by separating the prompts into global and local categories, enabling learning from distinct perspectives.
> Similarly, the decline in pixel-level PRO in the Medical domain seems to stem from the same issue. The addition of normal learnable tokens to the prompts likely reduced the tendency to identify anomalous regions. This can be attributed to the challenge of using a single anomaly prompt to simultaneously learn both global and local patterns.
> By separating the prompts into global and local categories, the global prompt effectively captures semantic anomalies from a high-level perspective, while the local prompt focuses on detecting fine-grained semantic anomalies. This separation has enabled a more effective application of the approach.

---

> ### Author Response · Authors · 2024-11-28
>
> > ### Q3. Explanation for F3 Performance Drop in Medical Applications.
>
> Thank you for your question regarding Table 3. F3 enhances the learning of global and local features through Global-Local Prompt Separation, allowing for more detailed representation of each. The loss function in this study places a stronger emphasis on pixel-level information. According to the dataset statistics provided in the Appendix A, the Medical domain suffers from a lack of normal data, which increases reliance on pixel-level loss during training.
> As a result, the simple separation of global and local prompts may not sufficiently adjust image-level features due to the absence of pixel-level information, leading to reduced performance. To address this issue, we introduced F4, Glocal Contrastive Learning, to complement the loss of global and local information and mitigate this limitation.

---

> ### Author Response · Authors · 2024-11-28
>
> > ### Q4. Explanation of Anchor Prompts.
>
> Thank you for your question regarding anchor prompts. As explained in General response 3, the global prompt was selected as the anchor based on the hierarchical characteristics of image semantics. This choice ensures better alignment of local text embeddings relative to global normality and abnormality. For a detailed explanation and supporting results, please refer to General response 3 and the ablation study summarized in the table. If further clarification is needed, we would be happy to provide additional details.

---

> ### Author Response · Authors · 2024-11-28
>
> > ### Q5. Clarification of Minor Issues
>
> Thank you for reading thoroughly and helping us address minor mistakes.
>
> Please let us know if further details are required. Thank you for your valuable feedback.

---

### Author Response · Authors · 2024-11-28
**General response (1/3)**

> ### Concern about the difference between AnomalyCLIP and Ours.

We would like to acknowledge and express our gratitude to the foundational works, including OpenCLIP [1], WinCLIP [2], PromptAD [3], AnomalyCLIP [4], which have greatly inspired this study. GlocalCLIP extends the concept of AnomalyCLIP by explicitly distinguishing between global and local information and employing Glocal Contrastive Learning to complementarily learn these two types of information.
This study highlights the critical role of prompts in decision-making for anomaly detection. By focusing on prompt learning, we aim to enhance the effectiveness of ZSAD through a complementary approach that integrates global and local perspectives.
In conclusion, we demonstrate the strength and effectiveness of our approach, which is both simple and effective, as evidenced by Tables 1, 2 and Fig. 3. We also extend our thanks to Reviewer 5eoh for recommending incremental experiments. Following your suggestion, we conducted ablation studies starting with AnomalyCLIP to clearly demonstrate the incremental improvements.

| Module | Industrial domain |  | Medical domain |  |
|:---:|:---:|:---:|:---:|:---:|
|  | **Pixel-level** | **Image-level** | **Pixel-level** | **Image-level** |
| **AnomalyCLIP** | (94.2, 81.0) | (86.1, 88.7) | (89.1, 72.0) | (94.4, 95.0) |
| **+Semantic design** | (95.0, 82.4) | (85.6, 88.3) | (90.0, 74.5) | (94.6, 95.2)   |
| **+Global local branch**| (95.3, **84.0**) | (86.2, 88.5) |(90.2, 74.4) | (89.8, 91.1) |
| **+GCL** | (**95.3**, 83.3) | (**86.7, 89.3**) | (**90.3, 74.8**) | (**94.9, 95.4**) |

We noted improvements in the Industrial domain, with Pixel-level AUROC increasing by 1.2%, AUPRO by 3.7%, Image-level AUROC by 0.7%, and AP by 1.4%. Similarly, in the Medical domain, Pixel-level AUROC increased by 1.3%, AUPRO by 3.9%, Image-level AUROC by 0.5%, and AP by 0.4%.

[References]

[1] Cherti, M., Beaumont, R., Wightman, R., Wortsman, M., Ilharco, G., Gordon, C., ... & Jitsev, J. (2023). Reproducible scaling laws for contrastive language-image learning. In Proceedings of the IEEE/CVF Conference on Computer Vision and Pattern Recognition (pp. 2818-2829).

[2] Jeong, J., Zou, Y., Kim, T., Zhang, D., Ravichandran, A., & Dabeer, O. (2023). Winclip: Zero-/few-shot anomaly classification and segmentation. In Proceedings of the IEEE/CVF Conference on Computer Vision and Pattern Recognition (pp. 19606-19616).

[3] Li, X., Zhang, Z., Tan, X., Chen, C., Qu, Y., Xie, Y., & Ma, L. (2024). Promptad: Learning prompts with only normal samples for few-shot anomaly detection. In Proceedings of the IEEE/CVF Conference on Computer Vision and Pattern Recognition (pp. 16838-16848).

[4] Zhou, Q., Pang, G., Tian, Y., He, S., & Chen, J. (2023). Anomalyclip: Object-agnostic prompt learning for zero-shot anomaly detection. arXiv preprint arXiv:2310.18961.

---

### Author Response · Authors · 2024-11-28
**General response (2/3)**

> ### Interpretation of the roles of global prompts and local prompts, as well as the differences between them.

The third row of Fig. 4 highlights the differences between global and local prompts. In the case of w/o GCL, the global and local prompts are independently trained. When the roles are switched, using the global prompt for pixel-level anomaly localization reveals limited performance in detecting localized anomalous regions. This serves as qualitative evidence demonstrating the distinct characteristics of global and local prompts. Specifically, the global prompt focuses on the overall semantics of an object, whereas the local prompt concentrates on identifying localized defects. Consequently, the two prompts learn to represent general normality and abnormality from different perspectives. Additionally, visualizations illustrating these differences can be found in Appendix Fig. 6 to Fig. 8.

---

### Author Response · Authors · 2024-11-28
**General response (3/3)**

> ### Explanation and Analysis of Anchor Prompts.

| Anchor prompt | Industrial domain |  | Medical domain |  |
|:---:|:---:|:---:|:---:|:---:|
|  | **Pixel-level** | **Image-level** | **Pixel-level** | **Image-level** |
| **Local prompt** | (95.1, 83.0) | (86.6, 89.0) | (90.2, 74.8) | (94.8, 95.3) |
| **Global prompt** | (**95.3, 83.3**) | (**86.7, 89.3**) | (**90.3, 74.8**) | (**94.9, 95.4**) |

The anchor prompt in Glocal Contrastive Learning is based on the global prompt. This design choice stems from the hierarchical characteristics of image semantics. While the global prompt captures the overall image context, the local prompt focuses on fine-grained details. Therefore, the global text embedding was selected as the anchor to guide the alignment of the local text embeddings.
Specifically, the anchor prompts were defined as the global normal prompt and the global anomaly prompt, enabling dual alignment that ensures the prompts are learned relative to the semantic context of global normality and abnormality. (Please refer to the updated Equations 2 and 3 for further details.)
We conducted an ablation study on the choice of anchor prompts. As shown in the table, using the global prompt as the anchor resulted in better prompt learning performance, validating the effectiveness of this design choice.

---

### Meta-Review · Area_Chair_wdqs · 2024-12-20

**Metareview:**

The paper proposes a zero shot anomaly detection method using CLIP features. There are several such methods and the contributions on the method lie in  using a dual-branch approach to separately model global and local anomaly semantics. These are then jointly optimized via global and local prompts. The proposed method is tested on 15 industrial and medical datasets and outperforms existing ZSAD methods, demonstrating its efficacy. Reviewers like the proposed dual branch approach and the fact that it is tested on multiple datasets. However, they all feel that  improvements over AnomalyCLIP is limited. The rebuttal did not seem to address the revivers concerns. Hence, given the ratings of the paper, it is not ready for acceptance at ICLR.

**Additional Comments On Reviewer Discussion:**

Reviewers were active with their responses to the authors. While reviewer fHMB did not have a response for the authors, the other 3 reviewers were active in their responses. 2/3 reviewers felt their concerns were not addressed adequately and while bsxX seem happy with the response, they were concerned if the paper met the quality bar of ICLR.

---

### Decision · Program_Chairs · 2025-01-22

Reject